# FlatLand: Personalized Graph Federated Learning via Tailored Lorentz Space

Jiahong Liu[1]   Ram Samarth B B[2]   Xinyu Fu[3]   Menglin Yang[4]   Weixi Zhang[3]   Rex Ying[2]   Irwin King[1]

## Abstract

Federated learning enables privacy-preserving collaborative training, but highly heterogeneous client data remain challenging, especially in graph federated learning where clients possess structurally diverse graphs. Existing personalized federated learning (PFL) methods ignore the intrinsic geometric properties of diverse graph structures. We propose FlatLand, a novel personalized **F**ederated **lea**rning method that embeds different clients' data in **t**ailored **L**orentz space of hyperbolic geometry. Our key insight is that hyperbolic geometry naturally accommodates the intrinsic negative curvature prevalent in real-world graphs, while the time-like dimension in Lorentz space provides a principled way to encode client-specific heterogeneity. We develop a parameter decoupling strategy that separates heterogeneous information (captured in time-like parameters) from common knowledge (preserved in space-like parameters), enabling direct aggregation without requiring client similarity estimation and extra calculation modules. Empirical results on diverse federated graph learning tasks demonstrate that FlatLand achieves superior performance, particularly in low-dimensional settings. Code is available in our GitHub repository.

## 1. Introduction

Federated learning (FL) has emerged as a paradigm that enables collaborative machine learning across multiple clients while preserving data privacy. Traditional FL struggles with

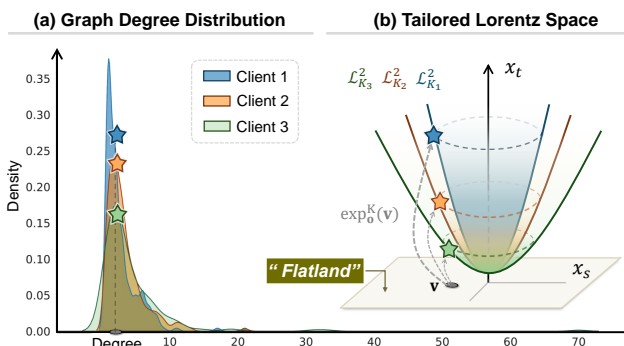

*Figure 1.* Example: (a) KDE of degree distributions from three CiteSeer clients (Davis et al., 2011), and (b) their respective 2D Lorentz Spaces with different Lorentz scale parameters $K$.

data heterogeneity, as one model cannot satisfy diverse local requirements (Tan et al., 2022). This challenge is magnified in graph federated learning, where complex topology yields pronounced structural heterogeneity across clients (Xie et al., 2021; Tan et al., 2023). In severe cases, federated learning may even underperform local training (Baek et al., 2023). To address heterogeneity, Personalized federated learning (PFL) resolves this by sharing common model knowledge and allowing for client-specific adaptations. Current PFL approaches for graph data primarily address heterogeneity through three main strategies during aggregation: (1) *parameter disentanglement*: splitting models into shared and personalized components (McMahan et al., 2017; Tan et al., 2023); (2) *client similarity estimation*: analyzing weights or gradients to evaluate client similarities (Xie et al., 2021); and (3) *auxiliary module calculation*: incorporating additional modules to distinguish between globally beneficial and client-specific parameters (Baek et al., 2023).

Despite their effectiveness, existing methods for PFL are confined to Euclidean space, implicitly assuming a uniform flat geometry across all client data distributions. This assumption necessitates the design of intricate mechanisms to address heterogeneity. Simple parameter disentanglement often fails, while more advanced techniques, such as client similarity estimation or auxiliary module integration, achieve better performance but incur significant computational overhead. Therefore, we revisit PFL through a geometric lens using Ricci curvature (Forman, 2003; Sun et al., 2024), which characterizes the intrinsic properties of

---

[1]Department of Computer Science and Engineering, The Chinese University of Hong Kong, Hong Kong SAR, China [2]Department of Computer Science, Yale University, New Haven, CT, USA [3]Huawei Technologies Co., Ltd., Hong Kong SAR, China [4]AI Thrust, The Hong Kong University of Science and Technology (Guangzhou), Guangzhou, Guangdong, China. Correspondence to: Jiahong Liu <jiahong.liu21@gmail.com>, Irwin King <king@cse.cuhk.edu.hk>.

*Proceedings of the 43ʳᵈ International Conference on Machine Learning*, Seoul, South Korea. PMLR 306, 2026. Copyright 2026 by the author(s).

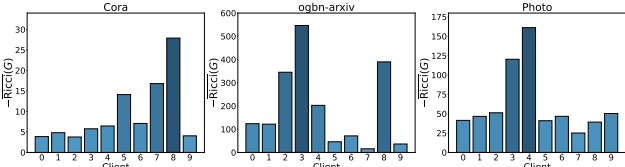

*Figure 2.* Averaged Forman-Ricci curvature across datasets (Cora, ogbn-arxiv, and Amazon-Photo). Higher bars indicate more pronounced non-Euclidean characteristics in these datasets.

graph structures: its sign indicates hyperbolic (negative), flat (zero), or spherical (positive) geometry, while its magnitude captures how much the space curves.

**Observations.** Our empirical analysis across multiple real-world datasets reveals two critical observations (Figure 3 and Figure 2): (1) client graphs predominantly exhibit *negative* Ricci curvature, indicating inherent hyperbolic structure, and (2) curvature values *vary substantially* across different clients, revealing intrinsic geometric heterogeneity that extends beyond simple statistical differences. These findings suggest that the assumption of unified Euclidean geometry in existing methods is fundamentally misaligned with the true geometric nature of graph data (Nickel & Kiela, 2018; Albert & Barabási, 2002; Khrulkov et al., 2020; Tan et al., 2023; He et al., 2025), resulting in suboptimal representations and complicating heterogeneity modeling.

To move beyond a single Euclidean geometry, we theoretically establish the advantages of **Lorentz geometry** for PFL (Section 4.1), showing it serves as a natural testbed for two reasons: **First**, enhanced representational power: Lorentz space enables low-distortion representation of the estimated inherent graph properties (Nickel & Kiela, 2018; Peng et al., 2021; Atigh et al., 2022; Yang et al., 2024). When client graphs exhibit varying Ricci curvature, assigning each client an appropriate hyperbolic curvature supports a more faithful modeling (Theorem 4.1). See Figure 1(a), distributions are long-tailed with varying skewness. In particular, Client 1 is 'steeper' and benefits from a Lorentz space with larger curvature magnitude (smaller $K$), which offers a 'roomier' embedding environment where tail nodes can be separated. **Second**, natural heterogeneity encoding: The additional time-like dimension in Lorentz space provides a carrier to capture intrinsic geometric heterogeneity across clients (Theorem 4.3). In the example (Figure 1(b))[1], heterogeneous properties such as "*how significant is the imbalance between tail nodes and head nodes?*" can be naturally distinguished in Lorentz space through the time-like dimension $x_t$, while common information (e.g., "*the star is a tail node*") remains preserved in the space-like dimensions $\mathbf{x}_s$ *("Flatland")* as a shared node representation.

---

[1]For convenience, all origins of Lorentz spaces in the figure are shown as the same, but in reality, their origins are not in the same location.

Based on the insights, we propose FlatLand, an exploratory PFL framework that embeds client data in tailored Lorentz spaces to faithfully capture their intrinsic geometry (Section 5). Yet such embedding alone cannot resolve the heterogeneity challenge in parameter aggregation. Therefore, we further leverage the nature of the time-like dimension and develop a theoretically grounded **parameter decoupling strategy** that designates heterogeneity-related parameters as personalized, while aggregating only those carrying shared information. This design effectively mitigates heterogeneity without auxiliary modules or client-similarity estimation and preserves the validity of Lorentz geometry.

To the best of our knowledge, this is the first work to bridge hyperbolic geometry and PFL for addressing client heterogeneity in a principled manner, providing a new **succinct** and **effective** perspective that leverages geometric properties. Experimental results demonstrate that FlatLand achieves superior performance than its Euclidean counterpart, particularly in low-dimensional settings that are crucial for communication-efficient federated learning.

## 2. Related Work

**Personalized federated learning on graphs.** Personalized federated learning (PFL) addresses statistical heterogeneity by learning client-adapted models instead of a single global model (Kairouz et al., 2021; Fallah et al., 2020; Jiang et al., 2019; Collins et al., 2021; Chen & Chao, 2022). For graph data, existing personalized federated graph learning methods commonly cluster clients by gradients (Xie et al., 2021), introduce additional personalized modules (Tan et al., 2023), or estimate client similarities for customized aggregation (Baek et al., 2023). These approaches are effective but usually operate in Euclidean spaces and rely on client-similarity estimation or auxiliary components to handle heterogeneity. Such designs do not explicitly model the scale-free and hierarchical structures widely observed in real-world graphs (Albert & Barabási, 2002; Krioukov et al., 2010), motivating a geometry-aware treatment of personalized graph FL.

**Hyperbolic federated learning.** Hyperbolic representations provide a natural geometry for hierarchical and power-law data (Nickel & Kiela, 2018; Chami et al., 2019). Recent federated methods leverage hyperbolic distances for knowledge distillation (An et al., 2023), hyperbolic prototypes for non-IID learning (Liao et al., 2023), or hyperbolic GNNs inside a FedAvg-style graph FL pipeline (Du et al., 2024). However, these methods do not personalize the underlying client geometry or separate client-specific geometric information from shared knowledge during aggregation. FlatLand differs by assigning each client a tailored Lorentz space and by decoupling time-like personalized parameters

from space-like shared parameters, enabling direct aggregation without client clustering or extra similarity estimation. A more detailed discussion is provided in Appendix A.

## 3. Preliminaries

**Lorentz Model of Hyperbolic Geometry.** The Lorentz model, also known as the hyperboloid model, is a representation of *Riemannian* hyperbolic space embedded in flat Minkowski ambient space $\mathbb{R}^{d+1}$ (Nickel & Kiela, 2018; Chen et al., 2021). Given a $d$-dimensional Lorentz manifold $\mathcal{L}_K^d$ with a constant negative curvature $-1/K(K > 0)$, suppose a point $\mathbf{x} \in \mathcal{L}_K^d$, which has the form $\mathbf{x} = \begin{bmatrix} x_t & \mathbf{x}_s \end{bmatrix}^\top \in \mathbb{R}^{d+1}$, where the first dimension $x_t \in \mathbb{R}$ is called *time-like* dimension and others $\mathbf{x}_s \in \mathbb{R}^d$ are *space-like* dimensions. It satisfies the following conditions: $\langle \mathbf{x}, \mathbf{x} \rangle_\mathcal{L} = -K$ and $x_t > 0$, where $\langle \mathbf{x}, \mathbf{y} \rangle_\mathcal{L} = -x_t y_t + \mathbf{x}_s^\top \mathbf{y}_s$ is the Lorentzian inner product. Here, $K$ is the positive Lorentz scale parameter; different $K$ values induce different hyperbolic geometries, with sectional curvature $-1/K$. Formal definitions are shown in Appendix B.1.

Typically, inputs reside in Euclidean space and need to be mapped into hyperbolic space. The way of projecting the data $\mathbf{v}^E \in \mathbb{R}^d$ in Euclidean to Lorentz space $\mathbf{x} \in \mathcal{L}_K^d$ can be simplified as [2]

$$
\begin{aligned}
\mathbf{x}^K &= \exp_\mathbf{o}^K(\mathbf{v}^E) = \exp_\mathbf{o}^K([0, \mathbf{v}^E]) = [x_t, \mathbf{x}_s]^\top, \\
x_t &= \sqrt{K} \cosh\left(\frac{\|\mathbf{v}^E\|_2}{\sqrt{K}}\right), \\
\mathbf{x}_s &= \sqrt{K} \sinh\left(\frac{\|\mathbf{v}^E\|_2}{\sqrt{K}}\right) \frac{\mathbf{v}^E}{\|\mathbf{v}^E\|_2}.
\end{aligned}
\tag{1}
$$

Different $K$ maps $\mathbf{v}^E$ to different Lorentz surfaces.

**Fully Lorentz Neural Networks.** Fully Lorentz network (Chen et al., 2021) has been shown to be ideal for PFL due to their reduced need for space projections, enhancing computational efficiency. These networks also incorporate Lorentz transformations (boosts and rotations), improving data heterogeneity handling and parameter interpretability (Appendix B.3).

Given an input vector $\mathbf{x} \in \mathcal{L}_K^n$, and a linear layer matrix $\mathbf{W} \in \mathbb{R}^{m \times (n+1)}$ to optimize, the fully Lorentz linear layer can be denoted as LT in a general form as $\text{LT}(\mathbf{x}; f; \mathbf{W}) := \left( \sqrt{\|f(\mathbf{W}\mathbf{x})\|^2 + K}, f(\mathbf{W}\mathbf{x}) \right)^T$, where $f$ is a function like activation, dropout, and bias.

---

[2]For clarity, all Lorentz space embeddings are denoted by $\cdot^H$. Specifically, if the Lorentz scale parameter is known as $K$, it is denoted by $\cdot^K$. In contrast, Euclidean space embeddings are denoted by $\cdot^E$. All vectors $\mathbf{x}$, if not superscripted, are assumed to be in Lorentz space.

**Problem Statement.** Given clients $\mathcal{C} = \{1, 2, \ldots, C\}$, each with a dataset $\mathcal{D}_c = (\mathbf{x}_i^c, y_i^c)_{i=1}^{N_c}$ and distribution $p_c(\mathbf{x}, y)$, Personalized Federated Learning (PFL) encounters heterogeneity if $p_i(\mathbf{x}, y) \neq p_j(\mathbf{x}, y)$ for any client pair $i \neq j$, which degrades performance. In PFL, the goal is to optimize personalized models $f_c(\cdot; \boldsymbol{\theta}_c, \boldsymbol{\theta}_s)$ for each client using specific and shared parameters $\boldsymbol{\theta}_c, \boldsymbol{\theta}_s$:

$$
\begin{aligned}
\min_{\boldsymbol{\theta}_c|_{c=1}^C, \boldsymbol{\theta}_s} \quad & \sum_{c=1}^C \mathbb{E}_{(\mathbf{x},y) \sim p_c(\mathbf{x},y)}[\mathcal{L}_c(f(\mathbf{x}; \boldsymbol{\theta}_c, \boldsymbol{\theta}_s), y)] \\
& + \lambda \Omega(\boldsymbol{\theta}_c|_{c=1}^C, \boldsymbol{\theta}_s).
\end{aligned}
\tag{2}
$$

This function merges local loss $\mathcal{L}_c$ with regularization $\Omega$, balanced by hyperparameter $\lambda$.

> **Our goals** are
>
> (1) to *effectively* represent the inherent properties of each local client data;
>
> (2) to *succinctly* reflect heterogeneity among client data and facilitate the communication of shared information without requiring additional computations.

## 4. Motivation and Insights

We investigate PFL for graph data from a geometric perspective via Ricci curvature (Appendix B.2), which distinguishes hyperbolic (negative), flat (zero), and spherical (positive) geometries and quantifies curvature strength through its magnitude. This provides a principled measure of the intrinsic geometry of client graphs, enabling us to analyze structural differences beyond conventional statistical heterogeneity. Figure 2 and Appendix C.1 show the empirical results on real-world datasets across clients, from which we identify **two consistent patterns**: (1) client graphs mostly have *negative* curvature, evidencing hyperbolic structure, and (2) curvature *values vary considerably* across clients, indicating geometric heterogeneity beyond statistical differences. These findings confirm that assuming a unified Euclidean geometry leads to distorted representations and ineffective heterogeneity modeling (Nickel & Kiela, 2018; Peng et al., 2021; Atigh et al., 2022), underscoring the necessity of exploring solutions beyond Euclidean space.

### 4.1. Motivation: Why Lorentz Space for PFL?

In this section, we bridge non-Euclidean geometry and PFL, and theoretically claim that the Lorentz geometry of hyperbolic space is particularly suitable, as it faithfully captures the non-Euclidean properties of client data and aligns with the goals outlined in Section 3.

**Why the Lorentz (hyperboloid) model specifically?**
While any model of hyperbolic space is geometrically equivalent, the Lorentz model provides unique structural advantages for PFL: (1) *Time–space decomposition*: The natural split into time-like and space-like coordinates directly supports our parameter decoupling strategy, where time-like parameters capture heterogeneity and space-like parameters are safely aggregated. (2) *Block-structured isometries*: Lorentz transformations have a linear-algebraic form that guarantees aggregated parameters remain on the hyperbolic manifold (Proposition 6.1). (3) *Closed-form curvature-aware gradients*: The exponential map yields analytically transparent, curvature-dependent gradient weighting. Other hyperbolic models (e.g., Poincaré ball) lack an intrinsically privileged direction for such clean decomposition.

### For Goal (1). Prevalent non-Euclidean heterogeneity can be captured by hyperbolic curvature.

The observed *negative* curvature shows that hyperbolic space naturally fits client graphs with such properties (Yang et al., 2022), and its curvature can be adjusted to accommodate *varied* client distributions (Krioukov et al., 2010). Next, we theorize the use of hyperbolic geometry in PFL.

**Theorem 4.1** (Necessity of tailored curvature). *Let $\{G_c\}_{c=1}^C$ be client graphs with average Forman-Ricci curvatures $\bar{R}_c = \overline{\mathrm{Ric}}(G_c)$, and let $\mathcal{L}_K^d$ denote the d-dimensional hyperbolic space with Lorentz scale parameter $K > 0$ and constant sectional curvature $-1/K < 0$. For each client $c$, let $\varepsilon_c^*(K)$ be the minimal edge distortion of any $(1+\varepsilon)$-bi-Lipschitz embedding $f_c : G_c \to \mathcal{L}_K^d$. Then the following holds:*

$$\max_{1 \le c \le C} \varepsilon_c^*(K) \ge \frac{c_d}{2} \max_{1 \le i < j \le C} |\bar{R}_i - \bar{R}_j|, \quad (3)$$

*where $c_d > 0$ is a dimension-dependent constant (Proof in Appendix D.1).*

> *Remark* 4.2. Theorem 4.1 indicates that if client graphs have different average Ricci curvatures, then no single Lorentz scale parameter $K$ can yield simultaneously small distortion for all clients; each client requires its own tailored curvature.

### For Goal (2). Strong correlation between heterogeneity and hyperbolic time-like dimension.

**Theorem 4.3.** *Let $C \in \{1, \ldots, m\}$, each client $c$ have Lorentz scale parameter $K_c > 0$ with $\mathrm{Var}(K_C) > 0$, and $\mathbf{x} = [x_t \ \mathbf{x}_s]^\top \in \mathbb{L}_{K_c}^d$ admit hyperbolic polar coordinates $(\rho, \mathbf{u})$ with $x_t = \sqrt{K_c} \cosh(\rho/\sqrt{K_c})$, $\mathbf{x}_s = \sqrt{K_c} \sinh(\rho/\sqrt{K_c}) \mathbf{u}$, $\mathbf{u} := \mathbf{x}_s/\|\mathbf{x}_s\|$. Then (1) $I(\mathbf{u}; C) = 0$; (2) $I(x_t; C) > 0$ if the pushforward measures of $T_K(\rho) := \sqrt{K} \cosh(\rho/\sqrt{K})$ differ across $K$; (3) $I((x_t, \mathbf{x}_s); C \mid K_c) = I(x_t; C \mid K_c)$ (Proof in Appendix D.2).*

> *Remark* 4.4. Theorem 4.3 shows that the time-like component (reflected in $\mathbf{x}_t$) retains mutual information with $C$. In other words, heterogeneity across clients is encoded along the time-like dimension, and the space-like part $\mathbf{x}_s$ carries no additional client-specific information.

## 4.2. Insights: Introduce a Higher Dimension (*time dimension*) to *"Flatland"*

In the above case, *"Flatland"* captures the common feature of a cylinder or a sphere, while a higher dimension (the third dimension) highlights the differences between the objects. Analogous to our setting, informally speaking, by introducing an additional *time-like* dimension, we can imagine each client's data residing in a unique Lorentz space (a curved world in a higher-dimensional space), where the curvature reflects the distinct distributions (objects). *"Flatland"*[3], $\mathbb{R}^d$ (flat), serves as a metaphor for a platform where common information (circle) is exchanged and integrated.

> *In "Flatland", a two-dimensional flat plane, the same shapes may represent the projections of various three-dimensional objects. For instance, a circle could be the projection of either a cylinder or a sphere from a higher dimension.*

## 5. The FlatLand Framework

We propose FlatLand, using tailored Lorentz spaces for each client and a well-designed parameter decoupling strategy to mitigate heterogeneity. The main steps are outlined in Figure 3 and Algorithm 2. Our method is succinct, directly built upon FedAvg, and requires no additional clustering computations or auxiliary modules; further details are provided in Appendix C.2.

### 5.1. Learnable Curvature Initialization

Each client is assigned a learnable Lorentz scale parameter $K_c$ (sectional curvature $-1/K_c$), rather than a fixed precomputed estimate. Client heterogeneity is not solely structural: node features, label distributions, and task-specific local optimization can also affect the preferred geometry, making any single pre-training statistic insufficient.

We use Forman-Ricci curvature only as a lightweight structure-aware initialization prior. Given client graph $G_c$, its average graph curvature initializes a raw scalar, which is

---

[3]Our method is named after Edwin Abbott's book "*Flatland: A Romance of Many Dimensions*", highlighting our insights of exploring a new perspective that maps various data distributions onto different Lorentz surfaces of hyperbolic geometry.

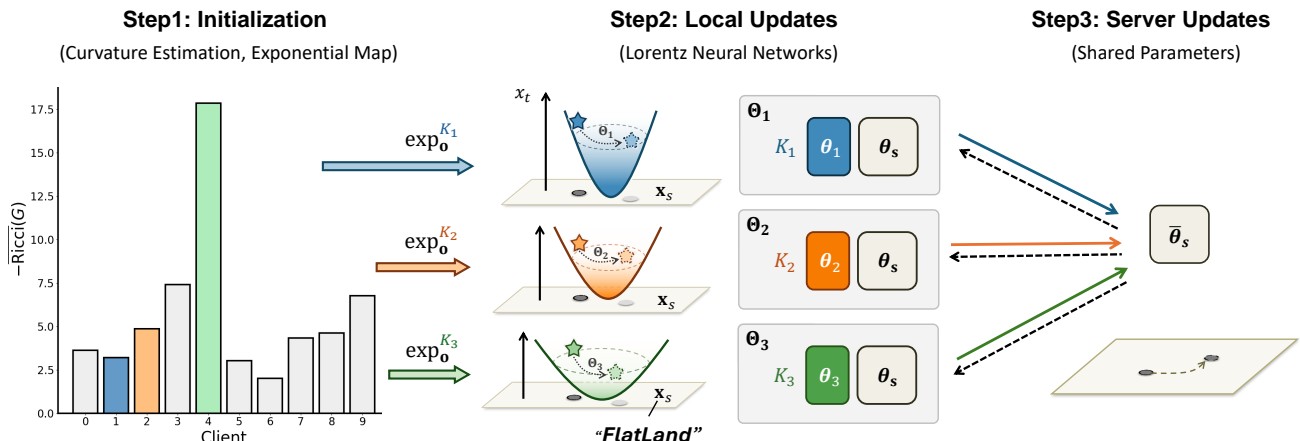

*Figure 3.* The FlatLand framework. It comprises three stages: (1) Initialization of the client-specific Lorentz scale (Section 5.1), personalized, and shared parameters; (2) Local updates within client-specific Lorentz spaces (Appendix C.2.3); and (3) Server updates, aggregating only shared parameters while preserving personalization by keeping personalized parameters locally (Section 5.2).

mapped to a positive $K_c$ by sigmoid reparameterization and updated during local training. This follows prior work treating curvature as a trainable geometric quantity rather than a fixed scaling constant (Gao et al., 2021; Ye et al., 2019; Gao et al., 2023). Details are provided in Appendix C.2.2.

## 5.2. Parameter Decoupling Strategy

Since each client has its own Lorentz space under the Lorentz model in Equation (1), the Lorentz scale parameter $K_c$ is kept client-specific. We further split the transformation parameters $\hat{\mathbf{M}}$ of the Lorentz linear layer into personalized parameters $\boldsymbol{\theta}_c$ and shared parameters $\boldsymbol{\theta}_s$. The split follows the Lorentz coordinates: $\boldsymbol{\theta}_s$ carries transferable information in *space-like* dimensions, while $\boldsymbol{\theta}_c$ captures client-specific heterogeneity through the *time-like* dimension.

For clarity, we derive the split on a Lorentz linear layer and omit the auxiliary functions $f$. Given input $\mathbf{x}^{(l)} = \begin{bmatrix} x_t^{(l)} & \mathbf{x}_s^{(l)} \end{bmatrix}^\top \in \mathcal{L}_K^n$ at layer $l$, where $x_t^{(l)} \in \mathbb{R}$ and $\mathbf{x}_s^{(l)} \in \mathbb{R}^n$. We decompose the learnable matrix $\hat{\mathbf{M}}^{(l)}$ as $\begin{bmatrix} \mathbf{m}^{(l)} & \mathbf{M}^{(l)} \end{bmatrix} \in \mathbb{R}^{m \times (n+1)}$, with $\mathbf{m}^{(l)} \in \mathbb{R}^{m \times 1}$ and $\mathbf{M}^{(l)} \in \mathbb{R}^{m \times n}$. Here, $\mathbf{m}^{(l)}$ controls the time-like contribution, whereas $\mathbf{M}^{(l)}$ operates on the transferable space-like dimensions. Let $\mathbf{z}^{(l)} = \mathbf{m}^{(l)} x_t^{(l)} + \mathbf{M}^{(l)} \mathbf{x}_s^{(l)} \in \mathbb{R}^m$. The output $\mathbf{x}^{(l+1)}$ is

$$\mathbf{x}^{(l+1)} = \text{LT}(\mathbf{x}^{(l)}; \hat{\mathbf{M}}^{(l)}) = \left( \sqrt{\|\mathbf{z}^{(l)}\|^2 + K}, \mathbf{z}^{(l)} \right)^T. \quad (4)$$

**Decoupled aggregation.** Guided by the derivation in Appendix C.3, we federate only the parameters associated with space-like dimensions. The reason is twofold. First,

Theorem 4.3 indicates that client-specific heterogeneity is expressed through the time-like coordinate. Since $\mathbf{m}^{(l)}$ directly weights $x_t^{(l)}$ in $\mathbf{z}^{(l)}$, averaging it across clients would mix scale-sensitive quantities from different Lorentz spaces. Second, $K_c$ determines the Lorentz surface where client $c$ embeds and updates its data, so it must remain local. After local training, client $c$ uploads only $\{\mathbf{M}_c^{(l)}\}_{l=1}^L$, and the server averages these space-like parameters. The averaged $\mathbf{M}^{(l)}$ is then combined locally with $\mathbf{m}^{(l)}$ and $K_c$. This does not assume identical space-like features across clients; rather, $\mathbf{M}$ acts on transferable space-like coordinates, while $\mathbf{m}^{(l)}$ and $K_c$ remain personalized. By Proposition 6.1, replacing $\mathbf{M}$ with its aggregated counterpart still maps outputs to the same client-specific Lorentz space $\mathcal{L}_{K_c}$.

> For the transformation parameters of a model $\mathcal{M}$ with $L$ layers:
> $$\boldsymbol{\theta}_c = \bigcup_{l=1}^{L} \{\mathbf{m}^{(l)}\}, \qquad \boldsymbol{\theta}_s = \bigcup_{l=1}^{L} \{\mathbf{M}^{(l)}\},$$
> where $\boldsymbol{\theta}_c$ and $\boldsymbol{\theta}_s$ denote the **personalized** and **shared** transformation parameter sets, respectively.

## 6. Analysis

This section analyzes three properties of FlatLand: **Correctness**, showing that client representations remain valid in Lorentz space during federated communication; **Convergence**, comparing its convergence behavior with FedAvg; and **Efficiency**, quantifying computational overhead.

**Correctness.** Although a fully Lorentz neural network ensures that representations remain in hyperbolic space during

*Table 1.* Comparison of node classification performance across real-world datasets with varying numbers of clients. The results, presented as mean and standard deviation, are based on five separate trials. Performances that are statistically significant ($p < 0.05$) are highlighted in bold.

| | Cora | | CiteSeer | | ogbn-arxiv | | Photo | |
|---|---|---|---|---|---|---|---|---|
| # clients | 10 | 20 | 10 | 20 | 10 | 20 | 10 | 20 |
| Local ($E$) | $79.94 \pm 0.24$ | $80.30 \pm 0.25$ | $67.82 \pm 0.13$ | $65.98 \pm 0.17$ | $64.92 \pm 0.09$ | $65.06 \pm 0.05$ | $91.80 \pm 0.02$ | $90.47 \pm 0.15$ |
| Local ($L$) | $78.35 \pm 0.05$ | $80.46 \pm 0.18$ | $72.30 \pm 0.04$ | $69.52 \pm 0.25$ | $65.85 \pm 0.09$ | $66.75 \pm 0.05$ | $91.76 \pm 0.10$ | $90.12 \pm 0.20$ |
| FedAvg | $69.19 \pm 0.67$ | $69.50 \pm 3.58$ | $63.61 \pm 3.59$ | $64.68 \pm 1.83$ | $64.44 \pm 0.10$ | $63.24 \pm 0.13$ | $83.15 \pm 3.71$ | $81.35 \pm 1.04$ |
| FedProx | $60.18 \pm 7.04$ | $48.22 \pm 6.81$ | $63.33 \pm 3.25$ | $64.85 \pm 1.35$ | $64.37 \pm 0.18$ | $63.03 \pm 0.04$ | $80.92 \pm 4.64$ | $82.32 \pm 0.29$ |
| FedPer | $79.35 \pm 0.04$ | $78.01 \pm 0.32$ | $70.53 \pm 0.28$ | $66.64 \pm 0.27$ | $64.99 \pm 0.18$ | $64.66 \pm 0.11$ | $91.76 \pm 0.23$ | $90.59 \pm 0.06$ |
| GCFL | $78.66 \pm 0.27$ | $79.21 \pm 0.70$ | $69.01 \pm 0.12$ | $66.33 \pm 0.05$ | $65.09 \pm 0.08$ | $65.08 \pm 0.04$ | $92.06 \pm 0.25$ | $90.79 \pm 0.17$ |
| FedGNN | $70.12 \pm 0.99$ | $70.10 \pm 3.52$ | $55.52 \pm 3.17$ | $52.23 \pm 6.00$ | $64.21 \pm 0.32$ | $63.80 \pm 0.05$ | $87.12 \pm 2.01$ | $81.00 \pm 4.48$ |
| FedSage+ | $69.05 \pm 1.59$ | $57.97 \pm 12.6$ | $65.63 \pm 3.10$ | $65.46 \pm 0.74$ | $64.52 \pm 0.14$ | $63.31 \pm 0.20$ | $76.81 \pm 8.24$ | $80.58 \pm 1.15$ |
| FED-PUB | $\mathbf{81.54 \pm 0.12}$ | $\underline{81.75 \pm 0.56}$ | $\underline{72.35 \pm 0.53}$ | $67.62 \pm 0.12$ | $\underline{66.58 \pm 0.08}$ | $\underline{66.64 \pm 0.12}$ | $\underline{92.73 \pm 0.18}$ | $\underline{91.92 \pm 0.12}$ |
| FedGTA | $\underline{80.59 \pm 0.38}$ | $79.01 \pm 0.31$ | $71.57 \pm 0.34$ | $69.94 \pm 0.14$ | $60.22 \pm 0.09$ | $58.74 \pm 0.14$ | $\mathbf{93.50 \pm 0.21}$ | $\mathbf{92.61 \pm 0.15}$ |
| AdaFGL | $80.09 \pm 0.00$ | $79.74 \pm 0.05$ | $72.34 \pm 0.00$ | $\underline{70.95 \pm 0.45}$ | $51.77 \pm 0.36$ | $50.94 \pm 0.08$ | $89.85 \pm 0.83$ | $88.11 \pm 0.05$ |
| FedHGCN | $72.09 \pm 0.16$ | $74.67 \pm 1.50$ | $66.98 \pm 0.56$ | $64.28 \pm 0.62$ | OOM | OOM | $79.26 \pm 0.56$ | $79.57 \pm 0.10$ |
| **FlatLand (Ours)** | $80.46 \pm 0.28$ | $\mathbf{82.49 \pm 0.25}$ | $\mathbf{73.90 \pm 0.23}$ | $\mathbf{72.24 \pm 0.24}$ | $\mathbf{67.52 \pm 0.16}$ | $\mathbf{67.64 \pm 0.04}$ | $92.49 \pm 0.19$ | $91.06 \pm 0.15$ |

*Table 2.* Comparison of node classification performance across heterophilic datasets with varying numbers of clients. The results, presented as mean and standard deviation, are based on five separate trials. Performances that are statistically significant ($p < 0.05$) are highlighted in bold.

| | Roman-empire | | Minesweeper | | Tolokers | | Questions | |
|---|---|---|---|---|---|---|---|---|
| # clients | 10 | 20 | 10 | 20 | 10 | 20 | 10 | 20 |
| Local ($E$) | $23.54 \pm 0.26$ | $23.70 \pm 0.32$ | $70.14 \pm 0.18$ | $66.78 \pm 0.11$ | $68.86 \pm 0.26$ | $62.764 \pm 0.69$ | $59.61 \pm 0.10$ | $60.40 \pm 0.21$ |
| Local ($L$) | $54.55 \pm 0.24$ | $49.54 \pm 0.35$ | $75.02 \pm 0.21$ | $70.71 \pm 0.14$ | $72.05 \pm 0.28$ | $70.35 \pm 0.70$ | $64.47 \pm 0.10$ | $62.68 \pm 0.21$ |
| FedAvg | $35.43 \pm 0.32$ | $32.00 \pm 0.39$ | $\underline{71.18 \pm 0.02}$ | $\underline{72.37 \pm 0.16}$ | $54.73 \pm 0.50$ | $56.36 \pm 0.39$ | $58.91 \pm 0.22$ | $60.33 \pm 0.15$ |
| FedProx | $26.43 \pm 1.41$ | $23.12 \pm 0.49$ | $70.66 \pm 0.20$ | $71.50 \pm 0.37$ | $41.15 \pm 0.22$ | $40.42 \pm 0.62$ | $45.46 \pm 0.34$ | $46.83 \pm 0.11$ |
| FedPer | $15.51 \pm 1.13$ | $15.45 \pm 2.76$ | $65.35 \pm 7.02$ | $53.80 \pm 11.40$ | $54.97 \pm 13.23$ | $44.82 \pm 11.61$ | $59.40 \pm 9.71$ | $62.32 \pm 1.56$ |
| GCFL | $29.44 \pm 0.49$ | $26.73 \pm 0.19$ | $71.14 \pm 0.09$ | $47.77 \pm 0.14$ | $19.81 \pm 0.57$ | $17.53 \pm 0.04$ | $45.71 \pm 0.25$ | $47.47 \pm 0.21$ |
| FedGNN | $29.09 \pm 0.01$ | $26.60 \pm 0.02$ | $71.12 \pm 0.09$ | $71.71 \pm 0.27$ | $41.57 \pm 0.07$ | $40.70 \pm 0.74$ | $45.73 \pm 0.26$ | $47.46 \pm 0.25$ |
| FedSage+ | $49.07 \pm 0.00$ | $38.36 \pm 0.00$ | $72.80 \pm 0.00$ | $69.70 \pm 0.00$ | $71.31 \pm 0.00$ | $\underline{69.73 \pm 0.00}$ | $\underline{65.06 \pm 0.00}$ | $59.33 \pm 0.00$ |
| FED-PUB | $36.77 \pm 0.30$ | $32.67 \pm 0.39$ | $71.56 \pm 0.05$ | $70.72 \pm 0.40$ | $\mathbf{72.46 \pm 0.68}$ | $65.26 \pm 0.59$ | $54.91 \pm 0.42$ | $62.48 \pm 2.92$ |
| FedGTA | $60.94 \pm 0.19$ | $59.65 \pm 0.28$ | $64.97 \pm 0.35$ | $49.63 \pm 8.64$ | $49.97 \pm 2.68$ | $50.68 \pm 3.94$ | $53.79 \pm 0.41$ | $61.70 \pm 0.35$ |
| AdaFGL | $\underline{64.55 \pm 0.00}$ | $\mathbf{62.42 \pm 0.26}$ | $65.59 \pm 0.56$ | $51.48 \pm 7.14$ | $49.82 \pm 2.17$ | $50.62 \pm 4.19$ | $54.87 \pm 0.52$ | $\underline{62.84 \pm 0.49}$ |
| FedHGCN | $55.99 \pm 0.18$ | $53.07 \pm 0.08$ | $66.37 \pm 0.05$ | $63.63 \pm 0.12$ | $65.69 \pm 0.05$ | $62.98 \pm 0.11$ | $43.21 \pm 0.08$ | $44.08 \pm 0.09$ |
| **FlatLand (Ours)** | $\mathbf{66.10 \pm 0.21}$ | $\underline{62.05 \pm 0.11}$ | $\mathbf{76.34 \pm 0.05}$ | $\mathbf{74.72 \pm 0.11}$ | $\underline{71.34 \pm 0.06}$ | $\mathbf{72.11 \pm 0.12}$ | $\mathbf{67.71 \pm 0.08}$ | $\mathbf{66.25 \pm 0.10}$ |

local training, as guaranteed by Lemma D.12, we further need to verify that our proposed decoupling strategy also preserves this property after server-side aggregation.

**Proposition 6.1.** *Let* $\hat{\mathbf{M}} = \begin{bmatrix} \mathbf{m} & \mathbf{M} \end{bmatrix}$, *where* $\hat{\mathbf{M}} \in \mathbb{R}^{m \times (n+1)}$, $\mathbf{m} \in \mathbb{R}^{m \times 1}$, *and* $\mathbf{M} \in \mathbb{R}^{m \times n}$. *Let* $\Phi\left(\hat{\mathbf{M}}, \mathbf{N}\right) = \begin{bmatrix} \mathbf{m} & \mathbf{N} \end{bmatrix}$, *where* $\mathbf{N} \in \mathbb{R}^{m \times n}$ *is the aggregated shared parameter matrix obtained from* $\{\mathbf{M}_i\}_{i=1}^{C}$ *using the proposed decoupling strategy. For all* $\mathbf{x} \in \mathcal{L}_K^n$, *we have* $\mathrm{LT}\left(\mathbf{x}; \Phi\left(\hat{\mathbf{M}}, \mathbf{N}\right)\right) \in \mathcal{L}_K^m$.

Proposition 6.1 (refer to the proof in Appendix D.3) implies that, even after the aggregation of shared parameters on the server, the transformation of any client vector $\mathbf{x} \in \mathcal{L}_K^n$ by this updated matrix will still yield results in the Lorentz space $\mathcal{L}_K^m$ with the same Lorentz scale parameter, and hence the same sectional curvature.

**Convergence.** We analyze in Appendix D.4 whether our method hinders the convergence rate. The results show that, under the standard FedAvg scheme, our method does not affect the convergence rate, which remains at $\mathcal{O}(1/T)$. The key reason is that FlatLand changes only the geometric parameterization of local updates. Server-side aggregation still averages shared space-like parameters as in FedAvg, without estimating client similarity.

**Efficiency.** We provide the time complexity analysis in Appendix C.4. FlatLand introduces minimal operations, like the $O(1)$ exponential map and curvature estimation, which can be mitigated by pre-computation. These minimal costs are offset by reduced communication overhead and enhanced representation in Lorentz space, making FlatLand efficient for practical personalized FL.

In summary, FlatLand preserves correctness by keeping representations in Lorentz space after aggregation, achieves the same convergence rate as FedAvg, and incurs only minimal overhead comparable to FedAvg. Besides, we further justify the rationale of our method from the perspective of Lorentz transformations in Appendix D.5.

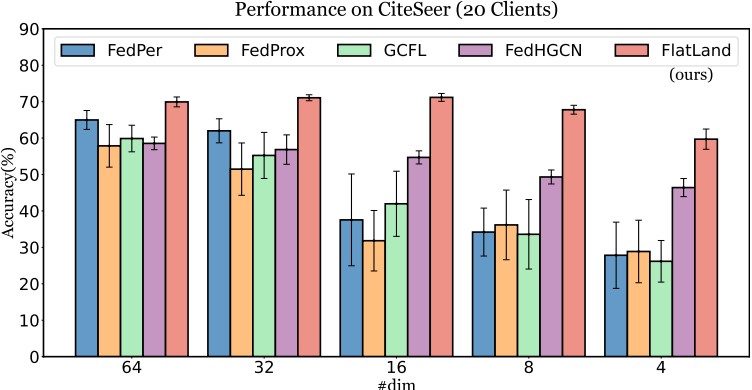

*Figure 4.* Performance of CiteSeer (20 clients) with varying dimensions for node classification scenario.

*Figure 5.* Ablation study of FlatLand on the Cora dataset.

## 7. Experiments

In this section, we validate the effectiveness of FlatLand through experiments on *node classification* and *graph classification* on a series of benchmark datasets. The experiments are designed to address the following research questions. **RQ1.** Can FlatLand outperform personalized and hyperbolic FL baselines? **RQ2.** Can FlatLand still perform well in low-dimensional settings? **RQ3.** Can FlatLand maintain high performance under partial client participation in FL? **RQ4.** Are the proposed novel components really beneficial?

### 7.1. Experimental Setup

**Datasets and Baselines.** The details about datasets are listed in Appendix E.1. Implementation details are shown in Appendix E.2. To assess FlatLand and demonstrate its superiority, we compare it with the following baselines: (1) Local: clients train their models locally without any communication. Local ($E$) refers to self-training in the Euclidean model, while Local ($L$) refers to training in the Lorentz model; (2) FedAvg (McMahan et al., 2017) and (3) Fed-Prox (Li et al., 2020a): the most popular FL baselines; (4) FedPer (Arivazhagan et al., 2019): a PFL baseline with personalized model layers; (5) GCFL (Xie et al., 2021): a PFGL baseline with client clustering and cluster-wise model aggregation; (6) FedGNN (Wu et al., 2021) and (7) FedSage+ (Zhang et al., 2021a): two FGL baselines; (8) FED-PUB (Baek et al., 2023): a PFGL baseline with personalized model aggregation and local weight masking; (9) FedGTA (Li et al., 2023) introduces a personalized optimization strategy that leverages topology-aware local smoothing confidence and mixed neighbor features; (10) AdaFGL (Li et al., 2024) addresses structural non-IID challenges by introducing a decoupled, two-stage personalized learning strategy ; (11) FedHGCN (Du et al., 2024): a hyperbolic FGL baseline that omits explicit client-geometry modeling. Together, these baselines cover local, standard FL, personalized FL, graph FL, and hyperbolic graph FL settings; further

selection rationale is provided in Appendix E.3.

### 7.2. Main Experimental Results (RQ1)

**Node Classification.** We tackle node classification on *highly heterogeneous homophilic and heterophilic datasets*, with non-overlapping node partitions for each client, which many previous methods are not designed to address. This challenge highlights our method's ability to handle heterogeneity that previous approaches could not address. Tables 1 and 2 show that our proposed FlatLand achieves the best or competitive performance in most settings, with particularly clear gains on CiteSeer, ogbn-arxiv, and Minesweeper. (1) Local ($L$) often surpasses Local ($E$), suggesting that hyperbolic space can better represent most datasets, with the difference being particularly pronounced in heterophilic graphs. (2) Vanilla Euclidean FL baselines such as FedAvg, FedProx, and FedGNN often underperform local training under strong heterogeneity. Among stronger Euclidean/PFGL baselines, FED-PUB is often the strongest on homophilic node datasets, while FedGTA is particularly competitive on Photo and FedSage+, FED-PUB, and AdaFGL remain competitive on some heterophilic datasets. (3) FedHGCN, despite operating in hyperbolic space, underperforms in many heterogeneous settings because it does not explicitly

*Table 3.* Performance on graph classification tasks. The results are reported as mean $\pm$ standard deviation over five runs. Bold indicates statistical significance ($p < 0.05$).

| | CHEM (1) | BIO-CHEM-SN (3) |
|---|---|---|
| # datasets | 7 | 13 |
| Local ($E$) | $75.54 \pm 1.73$ | $67.17 \pm 1.76$ |
| Local ($L$) | $75.72 \pm 2.41$ | $65.31 \pm 2.13$ |
| FedAvg | $75.88 \pm 2.17$ | $66.91 \pm 1.94$ |
| FedProx | $76.05 \pm 1.92$ | $66.34 \pm 2.26$ |
| FedPer | $75.81 \pm 2.17$ | $66.27 \pm 2.09$ |
| GCFL | $76.49 \pm 1.23$ | $67.21 \pm 2.39$ |
| FedHGCN | $75.06 \pm 1.81$ | OOM |
| **FlatLand (Ours)** | $\mathbf{76.55 \pm 2.28}$ | $\mathbf{67.31 \pm 2.58}$ |

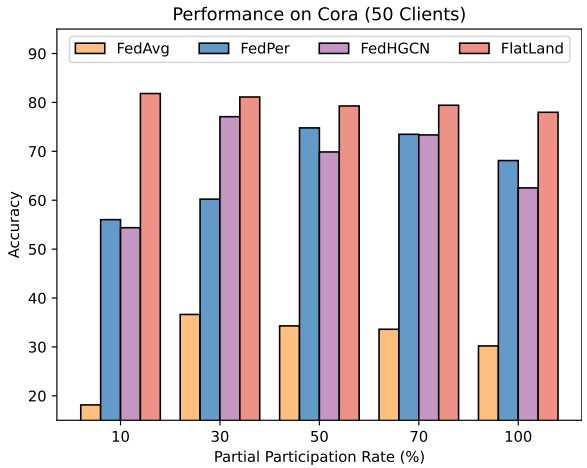

*Figure 6.* Performance comparison between FedAvg, FedPer, Fed-HGCN, and FlatLand under different client participation rates on Cora with 50 clients.

*Table 4.* Ablation study results about the necessity of using Lorentz space to do parameter decoupling.

|  | Cora (10) | Cora (20) | CiteSeer (10) | CiteSeer (20) |
|---|---|---|---|---|
| # datasets | 10 | 20 | 10 | 20 |
| FedAvg | $69.19 \pm 0.67$ | $69.50 \pm 3.58$ | $63.61 \pm 3.59$ | $64.68 \pm 1.83$ |
| FedPer | $\underline{79.35} \pm 0.04$ | $\underline{78.01} \pm 0.32$ | $70.53 \pm 0.28$ | $\underline{66.64} \pm 0.27$ |
| FlatLand ($E$) | $78.53 \pm 0.73$ | $76.23 \pm 0.43$ | $\underline{70.68} \pm 0.52$ | $66.29 \pm 0.35$ |
| FlatLand (**ours**) | $\mathbf{80.46} \pm 0.28$ | $\mathbf{82.49} \pm 0.25$ | $\mathbf{73.90} \pm 0.23$ | $\mathbf{72.24} \pm 0.24$ |

model client-specific geometry, akin to FedAvg vs Local ($E$) in Euclidean space. Due to the quadratic time and space complexity of FedHGCN's node selection module, it can easily encounter out-of-memory (OOM) issues with large datasets like ogbn-arxiv. In contrast, our experiments show that FlatLand effectively mitigates heterogeneity and yields substantial improvements on both highly heterogeneous homophilic datasets (e.g., CiteSeer) and heterophilic datasets (e.g., Minesweeper and Roman-Empire).

**Graph Classification.** Table 3 shows the results of the graph classification task, which is conducted with multiple datasets from one or more domains owned by different clients in each task/setting. In the single-dataset CHEM setting, Local ($L$) outperforms Local ($E$) due to inherent hyperbolic characteristics better captured by hyperbolic geometry. However, in multiple-dataset settings like BIO-CHEM-SN, Local ($L$) fails to surpass Local ($E$), potentially because not all datasets exhibit prominent hyperbolic features. With our proposed federated graph learning approach, FlatLand can significantly enhance the performance of the Lorentzian model, outperforming the Euclidean baselines, and demonstrating its effectiveness.

### 7.3. Varying Embedding Dimensions (RQ2)

Lower embedding and hidden dimensions reduce the parameter transmission cost in federated learning, as fewer parameters are communicated between the server and clients

during training. Considering the representational power of hyperbolic spaces in lower dimensions (Chami et al., 2019), we reduced the embedding dimension from 64 to 4 to evaluate FlatLand's ability to mitigate data heterogeneity using compact representations. Figure 4 shows the results on CiteSeer (20 clients), with similar trends observed across datasets. Dimensionality reduction from 64 to 4 had a relatively small impact on the hyperbolic methods (FlatLand and FedHGCN) compared to their Euclidean counterparts. Notably, while FedHGCN underperformed Euclidean methods at higher dimensions, it outperformed them when the dimension was reduced to 16. FlatLand consistently outperformed all other methods in different embedding dimensions, and its performance advantage over the baselines became increasingly significant as the dimensionality was reduced.

### 7.4. Partial Client Participation (RQ3)

We further evaluate whether FlatLand remains effective when only a subset of clients participates in each communication round. This setting is common in practical FL systems, where coordinating all clients simultaneously can be difficult. We conduct experiments on Cora with 50 clients, a larger client-pool configuration for graph FL (Du et al., 2024), and compare FedAvg, FedPer, FedHGCN, and FlatLand under different participation rates.

Figure 6 shows that FlatLand is robust to partial participation. Even with only 10% client participation (5 clients), FlatLand achieves 81.82% accuracy, while FedAvg reaches only 18.14%. Across all participation rates, FlatLand consistently outperforms all baselines, whereas FedAvg remains low and fluctuates because its aggregated model is sensitive to the sampled clients. These results suggest that separating personalized time-like parameters from shared space-like parameters reduces the damage caused by missing or inconsistent client-specific updates.

### 7.5. Ablation Study (RQ4)

To analyze the contribution of each component, we conduct ablation studies on the proposed design choices. A unified summary of these ablations is provided in Appendix E.4.

**The Benefits of Adaptive Curvature.** The "w/o TS" (without tailored curvature) in Figure 5 refers to setting a constant Lorentz scale parameter $K = 1$ for all clients instead of employing tailored curvature settings. The results indicate that using a fixed hyperbolic space with a constant Lorentz scale parameter yields inferior performance compared with adaptive tailored curvature. In contrast, learning client-specific Lorentz scale parameters allows the full FlatLand model to surpass the Local ($L$) baseline, showing that adaptive geometry is important for effective federated transfer.

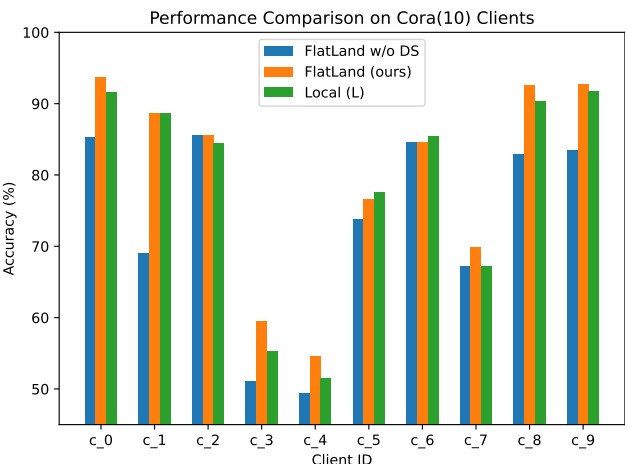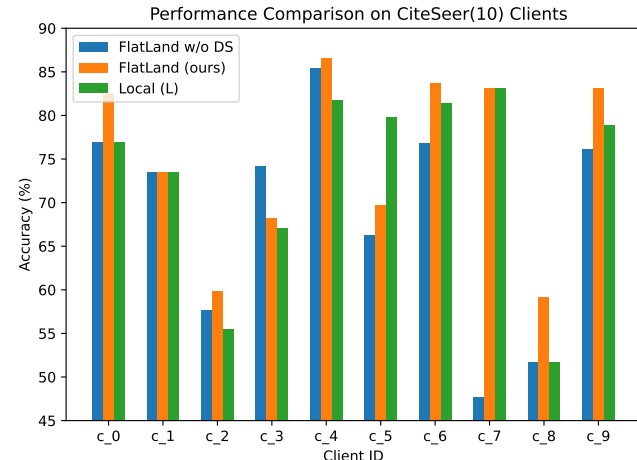

*Figure 7.* Client-level performance comparison on Cora (left) and CiteSeer (right) with 10 clients. Each group corresponds to one client and reports accuracy of FlatLand w/o DS, FlatLand, and Local ($L$). FlatLand improves most clients, while removing DS causes client-wise degradation, supporting the role of time-like parameter decoupling.

**The Benefits of Time-like Parameter Decoupling.** The "w/o DS" in Figure 5 refers to no parameter decoupling strategy (DS), which exhibits significant fluctuations across rounds because the aggregation process incorporates heterogeneous information, adversely affecting the results. This highlights the effectiveness of our proposed decoupling strategy and validates that the time-like dimension can effectively capture heterogeneous information. Moreover, we analyze the benefits of DS for each client's performance. As shown in Figure 7, with client IDs on the x-axis, Flatland outperforms the local method for the vast majority of clients, notably improving performance for clients with inherently poorer results, like c_8 in the CiteSeer dataset. This underscores *the necessity of federated settings for hyperbolic models.* Without our proposed DS, performance deteriorates significantly (e.g., c_7 in CiteSeer), further *validating our hypothesis that the time-like parameter encapsulates crucial heterogeneity information.*

**The Necessity of Lorentz Space.** We conducted experiments to further evaluate the necessity of using Lorentz space. Table 4 presents the results of an ablation study on the Lorentz transformation. FlatLand ($E$) represents our proposed method with a parameter decoupling strategy implemented using a Euclidean backbone. Without Lorentz geometry, FlatLand ($E$) underperforms because the time-like parameter loses its geometric meaning. It even falls short of FedPer in most cases, which uses the classifier layer for personalization. These results validate our hypothesis and underscore the importance of hyperbolic representation.

**Robustness of Curvature Initialization.** Appendix E.5 shows that FlatLand remains stable across Ricci, Ollivier, constant, and MLP initialization strategies, indicating that learning client-specific curvature matters more than the particular initialization rule. This suggests that curvature ini-

tialization mainly provides a warm start, while the learnable $K_c$ can adapt to each client's geometry during training.

# 8. Conclusions

This paper introduces FlatLand, an exploratory PFL approach that uses hyperbolic geometry to model heterogeneous client graph distributions. By assigning clients tailored Lorentz spaces and learning client-specific Lorentz scale parameters, FlatLand preserves local geometric structure while avoiding explicit client similarity estimation. Its parameter decoupling strategy separates personalized time-like parameters from shared space-like parameters, allowing the server to aggregate common information while reducing interference from heterogeneous updates. Experiments on node- and graph-level benchmarks show that this geometric design improves personalization, particularly when compact representations are required, and highlights non-Euclidean geometry as a promising direction for federated personalization in heterogeneous settings.

**Limitation and future work.** Hyperbolic geometry is not universally optimal for all graph structures, since some clients may be closer to Euclidean or positively curved geometries. Future work can extend FlatLand to mixed-curvature spaces and evaluate richer Lorentz backbones and broader data modalities beyond graph benchmarks.

# Acknowledgments

Jiahong Liu and Irwin King acknowledge partial support from RGC of Hong Kong SAR, China (CUHK 2300246, RGC C1043-24G; CUHK 14203425, RGC GRF 2151317). Menglin Yang acknowledges partial support from the General Program of Guangdong Provincial Natural Science Foundation (No. 2026A1515012118).

## Impact Statement

This work presents a novel geometric approach to personalized federated learning that enhances privacy-preserving collaborative machine learning. The proposed FlatLand framework inherently supports privacy by design through local training without requiring raw data sharing. Compared to many PFL methods that additionally share similarity matrices, clustering assignments, or other client-level statistics, FlatLand is more privacy-preserving: only the shared (space-like) parameters are sent to the server for aggregation, while personalized parameters and Lorentz scale parameters remain local to each client. Standard privacy-enhancing techniques such as secure aggregation or differential privacy can be applied on top of FlatLand in the same way as for standard FL methods. We evaluate our approach on standard academic benchmarks without sensitive personal information.

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

# Appendix

## Contents

# A. Related Work

**Personalized Federated Learning**    Under statistical heterogeneity (Kairouz et al., 2021), conventional FL frameworks such as FedAvg (McMahan et al., 2017) often fail to obtain a single global model that generalizes well to every client (the basic framework is shown in Appendix B.4). Motivated by this, researchers have proposed personalized FL (PFL) to train customized local models (Tan et al., 2022; Liu et al., 2025; 2026). Generally speaking, existing PFL techniques can be categorized into the following three groups: (1) techniques that personalize client models via local fine-tuning (Fallah et al., 2020; Jiang et al., 2019; Wang et al., 2019), (2) techniques that personalize client models via customized model aggregation (Huang et al., 2021; Li et al., 2021b; Luo & Wu, 2022; Sun et al., 2021; Zhang et al., 2023; 2021b), and (3) techniques that personalize client models by creating localized models or layers (Arivazhagan et al., 2019; Chen & Chao, 2022; Collins et al., 2021; Deng et al., 2020; Dinh et al., 2020; Hanzely & Richtárik, 2020; Li et al., 2021a; Mansour et al., 2020). However, these PFL methods typically encode data samples in Euclidean spaces, which makes it difficult to capture scale-free properties and implicit hierarchical structure in client data.

**Personalized Federated Graph Learning**    When applied to graph data, personalized federated graph learning (PFGL) can intuitively exhibit the problem mentioned above. For example, (Xie et al., 2021) clusters clients based on gradients to aggregate models with similar data distributions. Another method (Tan et al., 2023) introduces additional personalized models to capture client-specific knowledge of graph structure. (Baek et al., 2023) calculates client-client similarities to apply personalized model aggregation with local weight masking. These methods learn node representations in Euclidean spaces, which cannot naturally model the power-law degree distributions that widely exist in real-world graph data (Albert & Barabási, 2002; Krioukov et al., 2010). More broadly, graph-structure mining and hyperbolic graph representation learning show that structural patterns and non-Euclidean geometry are central to graph modeling (Liu et al., 2022; 2021; Zhang et al., 2025; Fu et al., 2026). Additionally, the client clustering procedure and additional model components introduce computational overhead that may not be feasible in real-world scenarios with strict privacy constraints or limited resources.

**Hyperbolic Federated Learning**    Only a few studies have considered incorporating hyperbolic spaces into federated settings. (An et al., 2023) leverages hyperbolic distances to distill knowledge from the global model to the local model, to mitigate model inconsistency caused by data heterogeneity. (Liao et al., 2023) applies hyperbolic prototype learning to capture the hierarchical structure among data samples. As the work most similar to our FlatLand, FedHGCN (Du et al., 2024) is a simple combination of FedAvg and hyperbolic graph neural networks along with a node selection process. Although these methods can benefit from the hyperbolic space to capture the hierarchical structure in the data, they do not have the personalization capability to adaptively model client data spaces with different curvatures. This may lead to suboptimal results when there is severe data heterogeneity. Therefore, our goal is to design a novel FL framework that can encode client data in hyperbolic spaces with adaptive curvatures using personalization techniques.

# B. Preliminaries

## B.1. Lorentz (Hyperboloid) Model: Formal Definitions

Hyperbolic space is non-Euclidean geometry with a constant negative curvature. The curvature of hyperbolic space is a measure of how the geometry of the space deviates from the flatness of Euclidean space. The Lorentz manifold, also known as the hyperboloid model, is one of the most commonly used mathematical representations of hyperbolic space. Its greater stability for numerical optimization makes it a popular choice for hyperbolic geometry methods (Nickel & Kiela, 2018).

**Clarification on terminology.** Throughout this paper, "Lorentz space" or "Lorentz model" refers specifically to the hyperboloid model of *Riemannian* hyperbolic space, not to Lorentzian spacetime in the sense of pseudo-Riemannian geometry or general relativity. We use the Lorentzian inner product purely as a mathematical tool for representing hyperbolic geometry, consistent with prior hyperbolic learning literature (Nickel & Kiela, 2018; Chen et al., 2021; Zhang et al., 2021c). All constructions take place in flat Minkowski ambient space $\mathbb{R}^{d+1}$, where the Lorentz group $O(1, d)$ acts as global isometries. We do not assume curved Lorentzian manifolds (e.g., anti-de Sitter space), and notions such as light cones, causality, or speed limits from physics are not invoked or interpreted in our framework.

**Definition B.1** (Lorentz Manifold). A $d$-dimensional Lorentz manifold $\mathcal{L}_K^d$ with constant negative curvature $-1/K$ ($K > 0$) is defined as the hyperboloid $\mathbb{H}_K^d = \left\{ \mathbf{x} \in \mathbb{R}^{d+1} : \langle \mathbf{x}, \mathbf{x} \rangle_{\mathcal{L}} = -K, x_0 > 0 \right\}$ endowed with the Riemannian metric induced by the Lorentzian inner product.

**Definition B.2** (Lorentzian Inner Product). The inner product $\langle \mathbf{x}, \mathbf{y} \rangle_{\mathcal{L}}$ for $\mathbf{x}, \mathbf{y} \in \mathbb{R}^{d+1}$ is defined as $\langle \mathbf{x}, \mathbf{y} \rangle_{\mathcal{L}} = -x_0 y_0 + \sum_{i=1}^d x_i y_i$.

Based on the constraint $\langle \mathbf{x}, \mathbf{x} \rangle_{\mathcal{L}} = -K$, it holds for any point $\mathbf{x} = (x_0, \mathbf{x}') \in \mathbb{R}^{d+1}$ that $\mathbf{x} \in \mathcal{L}_K^d \Leftrightarrow x_0 = \sqrt{\|\mathbf{x}'\|_2^2 + K}$. Here, $K$ is the positive Lorentz scale parameter, and the corresponding sectional curvature is $-1/K$.

Next, the corresponding Lorentzian distance function for two points $\mathbf{x}, \mathbf{y} \in \mathcal{L}_K^d$ is provided as

$$d_{\mathcal{L}}^K(\mathbf{x}, \mathbf{y}) = \sqrt{K} \operatorname{arcosh}(-\langle \mathbf{x}, \mathbf{y} \rangle_{\mathcal{L}} / K). \tag{5}$$

**Definition B.3** (Tangent Space). For a point $\mathbf{x} \in \mathcal{L}_K^d$, the tangent space $\mathcal{T}_{\mathbf{x}} \mathcal{L}_K^d$ consists of all vectors orthogonal to $\mathbf{x}$, where orthogonality is defined with respect to the Lorentzian inner product (Definition B.2). Hence, $\mathcal{T}_{\mathbf{x}} \mathcal{L}_K^d = \{ \mathbf{v} : \langle \mathbf{x}, \mathbf{v} \rangle_{\mathcal{L}} = 0 \}$.

**Definition B.4** (Exponential and Logarithmic Maps). Let $\mathbf{v} \in \mathcal{T}_x \mathcal{L}_K^d$. The exponential map $\exp_{\mathbf{x}}^K : \mathcal{T}_{\mathbf{x}} \mathcal{L}_K^d \to \mathcal{L}_K^d$ and logarithmic map $\log_{\mathbf{x}}^K : \mathcal{L}_K^d \to \mathcal{T}_{\mathbf{x}} \mathcal{L}_K^d$ are defined as

$$\exp_{\mathbf{x}}^K(\mathbf{v}) = \cosh \left( \frac{\|\mathbf{v}\|_{\mathcal{L}}}{\sqrt{K}} \right) \mathbf{x} + \sqrt{K} \sinh \left( \frac{\|\mathbf{v}\|_{\mathcal{L}}}{\sqrt{K}} \right) \frac{\mathbf{v}}{\|\mathbf{v}\|_{\mathcal{L}}},$$

$$\log_{\mathbf{x}}^K(\mathbf{y}) = d_{\mathcal{L}}^K(\mathbf{x}, \mathbf{y}) \frac{\mathbf{y} + \frac{1}{K} \langle \mathbf{x}, \mathbf{y} \rangle_{\mathcal{L}} \mathbf{x}}{\left\| \mathbf{y} + \frac{1}{K} \langle \mathbf{x}, \mathbf{y} \rangle_{\mathcal{L}} \mathbf{x} \right\|_{\mathcal{L}}},$$

where $\|\mathbf{v}\|_{\mathcal{L}} = \sqrt{\langle \mathbf{v}, \mathbf{v} \rangle_{\mathcal{L}}}$ denotes the norm of $\mathbf{v}$ in $\mathcal{T}_{\mathbf{x}} \mathcal{L}_K^d$.

Particularly, for the sake of calculation, the origin of Lorentz manifold $\mathbf{o} = (\sqrt{K}, 0, 0, \ldots, 0) \in \mathcal{L}_K^d$ is chosen as the reference point for the exponential and logarithmic maps, which can be simplified as

$$\exp_{\mathbf{o}}^K(\mathbf{v}) = \exp_{\mathbf{o}}^K \left( \left[ 0, \mathbf{v}^E \right] \right)$$

$$= \left( \underbrace{\sqrt{K} \cosh \left( \frac{\|\mathbf{v}^E\|_2}{\sqrt{K}} \right)}_{\text{time-like dimension}}, \underbrace{\sqrt{K} \sinh \left( \frac{\|\mathbf{v}^E\|_2}{\sqrt{K}} \right) \frac{\mathbf{v}^E}{\|\mathbf{v}^E\|_2}}_{\text{space-like dimension}} \right), \tag{6}$$

where the $(,)$ denotes concatenation and the $\cdot^E$ denotes the embedding in Euclidean space .

## B.2. Forman-Ricci Curvature

Curvature is a metric used in Riemannian geometry that expresses how far a curved line deviates from a straight line, or how much a surface deviates from planarity. In this context, knowledge of the local and global geometrical features depends on an understanding of sectional curvature and Ricci curvature, respectively (Sun et al., 2024; Ye et al., 2019).

**Sectional Curvature.** This type of curvature is determined at any given point on a manifold by examining all possible two-dimensional subspaces that intersect at that point. It provides a more straightforward representation than the Riemann curvature tensor (Lee, 2018). Recent studies (Chen et al., 2021) often treat sectional curvature uniformly across the manifold, simplifying it to a singular constant value.

**Ricci Curvature.** Ricci curvature averages the sectional curvatures at a specific point. In graph theory, various discrete versions of Ricci curvature have been developed, such as Ollivier-Ricci curvature (Ollivier, 2009) and Forman-Ricci curvature (Forman, 2003). The Ricci curvature on graphs is intended to assess how the local structure around a graph edge deviates from that of a grid graph. Notably, the Ollivier approach provides a rougher estimate of Ricci curvature, whereas the Forman method is more combinatorial and computationally efficient.

For a weighted graph $G = (V, E, w)$, the overall Forman-Ricci curvature $\overline{\mathrm{Ric}}(G)$ can be calculated as follows:

$$\overline{\mathrm{Ric}}(G) = \frac{1}{|E|} \sum_{(i,j) \in E} \mathrm{Ric}(i,j),$$

where $|E|$ represents the cardinality of the edge set $E$ (i.e., the total number of edges), and $\mathrm{Ric}(i,j)$ is the Forman-Ricci curvature of the edge $(i, j)$, computed as (Southern et al., 2023)

$$\mathrm{Ric}(i,j) =: w_e \left( \frac{w_i}{w_e} + \frac{w_j}{w_e} - \sum_{e_l \sim i} \frac{w_i}{\sqrt{w_e w_{e_l}}} - \sum_{e_l \sim j} \frac{w_j}{\sqrt{w_e w_{e_l}}} \right)$$

where $w_e$ denotes the weight of the edge $e$, i.e., $(i, j)$, and $w_i$ and $w_j$ are the weights of vertices $i$ and $j$, respectively. The sums over $e_l \sim k$ run over all edges $e_l$ incident on vertex $k$ excluding $e$. Specifically, when vertex and edge weights are set to 1, the curvature is

$$\mathrm{Ric}(i,j) := 4 - d_i - d_j + 3|\#\Delta|,$$

where $d_i$ is the degree of node $i$ and $|\#\Delta|$ is the number of 3-cycles (i.e. triangles) containing the adjacent nodes.

In our experiments, Forman-Ricci curvature is computed once before training using the `GraphRicciCurvature` package. We do not introduce additional approximations beyond the package implementation; therefore, this curvature-estimation step is a preprocessing cost and is not part of the per-round federated training loop.

Therefore, the overall Forman-Ricci curvature of the graph is the weighted average of the curvature values of all edges.

### B.3. Lorentz Transformations (Hyperbolic Isometries)

In the context of hyperbolic geometry, Lorentz transformations are isometries of the hyperboloid model that preserve the Lorentzian inner product. While the terminology originates from special relativity, in our setting these transformations serve purely as mathematical tools for manipulating hyperbolic embeddings (Chen et al., 2021; Zhang et al., 2021c). They can be decomposed into a combination of a Lorentz Boost and a Lorentz Rotation (Moretti, 2002). The Lorentz boost is parameterized by a vector $v \in \mathbb{R}^n$ with $\|v\| < 1$ and represented by the matrix $B$. The Lorentz rotation matrix $R$ rotates space-like coordinates and is a special orthogonal matrix, i.e., $R^\top R = I$ and $\det(R) = 1$.

**Definition B.5** (Lorentz Boost). A Lorentz boost is a standard isometry of the hyperboloid model. Given a vector $\mathbf{v} \in \mathbb{R}^n$ with $\|\mathbf{v}\| < 1$ and the factor $\gamma = \frac{1}{\sqrt{1 - \|\mathbf{v}\|^2}}$, the Lorentz boost matrix is defined as:

$$\mathbf{B} = \begin{bmatrix} \gamma & -\gamma \mathbf{v}^\top \\ -\gamma \mathbf{v} & \mathbf{I} + \frac{\gamma^2}{1+\gamma} \mathbf{v}\mathbf{v}^\top \end{bmatrix} \tag{7}$$

where $\mathbf{I}$ is the $n \times n$ identity matrix.

In our use, this matrix is interpreted only as a hyperbolic isometry acting on embeddings.

**Definition B.6** (Lorentz Rotation). A Lorentz rotation describes a rotation of the spatial coordinates. The Lorentz rotation matrix is defined as:

$$\mathbf{R} = \begin{bmatrix} 1 & \mathbf{0}^\top \\ \mathbf{0} & \tilde{\mathbf{R}} \end{bmatrix} \tag{8}$$

where $\tilde{\mathbf{R}} \in \mathrm{SO}(n)$ is a special orthogonal matrix satisfying $\tilde{\mathbf{R}}^\top \tilde{\mathbf{R}} = \mathbf{I}$ and $\det(\tilde{\mathbf{R}}) = 1$.

A Lorentz rotation changes the orientation of the space-like coordinates while leaving the time-like coordinate unchanged.

Both the Lorentz boost and the Lorentz rotation are linear transformations defined directly in the Lorentz model. For any point $\mathbf{x} \in \mathcal{L}_K^n$, we have $\mathbf{B}\mathbf{x} \in \mathcal{L}_K^n$ and $\mathbf{R}\mathbf{x} \in \mathcal{L}_K^n$.

### B.4. The FedAvg Algorithm

Federated Learning (FL) is a distributed learning approach that enables the training of machine learning models using data residing on local devices. A cornerstone algorithm within the FL paradigm is the FedAvg algorithm (McMahan et al., 2017). FedAvg is particularly effective for scenarios where data is decentralized and not identically distributed across participants.

---

**Algorithm 1** FedAvg

---

**Require:** Model parameters $\boldsymbol{\theta}$, learning rate $\eta$, and client dataset $\mathcal{D}_c$ for each client $c \in \mathcal{C}$
**Ensure:** Aggregated model parameters $\boldsymbol{\theta}$
1: Initialize model parameters $\boldsymbol{\theta}^{(0)}$
2: **for** each communication round $r$ **do**
3:     **for** each client $c$ in $\mathcal{C}$ **do**
4:         Client $c$ receives global model parameters $\boldsymbol{\theta}^{(r)}$
5:         **for** local epochs $e$ **do**
6:             Compute gradients $\nabla \mathcal{L} = \nabla_{\boldsymbol{\theta}^{(r)}} \sum_{(\mathbf{x},\mathbf{y}) \in \mathcal{D}_c} \mathcal{L}_c(f(\mathbf{x}; \boldsymbol{\theta}^{(r)}), y)$
7:         **end for**
8:         Update local model $\boldsymbol{\theta}^{(r+1)} \leftarrow \boldsymbol{\theta}^{(r)} - \eta \nabla \mathcal{L}$
9:         Send $\boldsymbol{\theta}^{(r+1)}$ to the server
10:     **end for**
11:     $N = \sum_{c \in \mathcal{C}} |\mathcal{D}_c|$
12:     Server aggregates models $\boldsymbol{\theta}^{(r+1)} \leftarrow \sum_{c \in \mathcal{C}} \frac{|\mathcal{D}_c|}{N} \boldsymbol{\theta}_c^{(r+1)}$
13: **end for**

---

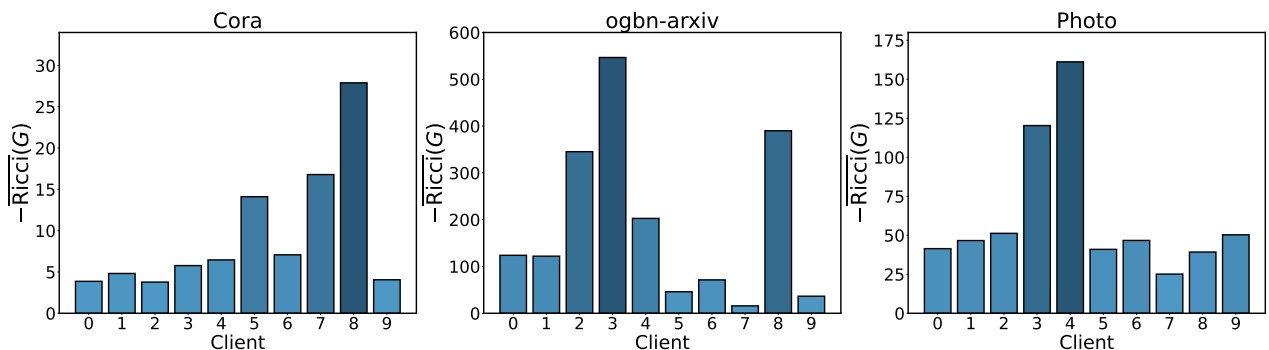

*Figure 8.* Averaged Forman-Ricci curvature across datasets (Cora, ogbn-arxiv, and Amazon-Photo). Higher bars indicate more pronounced non-Euclidean characteristics in these datasets.

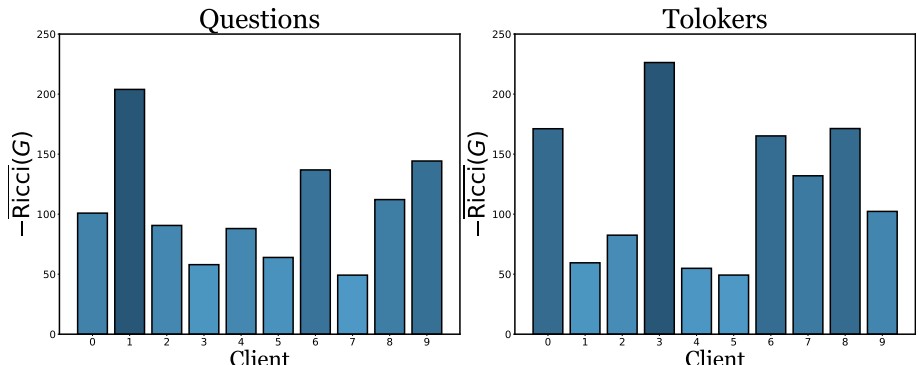

*Figure 9.* Averaged Forman-Ricci curvature across heterophilic datasets (Questions and Tolokers). Higher bars indicate more pronounced non-Euclidean characteristics in these datasets.

## C. Methodology Supplementary

### C.1. Statistics of Forman-Ricci Curvature in Other Datasets

We compute the Forman-Ricci curvature (Appendix B.2) for each client in the Cora, Photo, and ogbn-arxiv datasets, with 10 clients per dataset. The CiteSeer client statistics are shown in the initialization stage of Figure 3.

### C.2. The FlatLand Algorithm

This section introduces the supplementary details of our FlatLand with pseudocode shown in Algorithm 2.

#### C.2.1. OVERALL PROCESS

**S1 Initialization.** At the initial communication round $r = 0$, the parameters that need to be initialized can be divided into three groups:

(1) Lorentz scale parameters of $C$ clients $\{K_1, K_2, \ldots, K_C\}$;
(Section 5.1)

(2) Personalized parameters of $C$ clients $\{\boldsymbol{\theta}_1, \boldsymbol{\theta}_2, \ldots, \boldsymbol{\theta}_C\}$;                                                     (Section 5.2)

(3) Shared parameters $\overline{\boldsymbol{\theta}}_s$ of the central server.

All the parameters of client $i$ at round 0 can be written as $\boldsymbol{\Theta}_i^{(0)} = \left( K_i; \boldsymbol{\theta}_i^{(0)}; \overline{\boldsymbol{\theta}}_s^{(0)} \right)$ and server parameters as $\overline{\boldsymbol{\theta}}_s^{(0)}$.

**S2 Local updates.** Given the learning rate $\eta$ for the round $r$, each local client model performs training on the data $\mathcal{D}_i$ to minimize the task loss $\mathcal{L}(\mathcal{D}_i; \boldsymbol{\Theta}_i^{(r)})$ and then updates the parameters $\boldsymbol{\Theta}_i^{(r+1)} \leftarrow \boldsymbol{\Theta}_i^{(r)} - \eta \nabla \mathcal{L}$.     (Appendix C.2.3)

**S3 Server updates.** After local training, only the shared parameters $\boldsymbol{\theta}_{s_c}^{(r+1)}$ are updated on the server for each client

$c$. These are then aggregated using FedAvg: $\overline{\boldsymbol{\theta}}_s^{(r+1)} \leftarrow \sum_{c=1}^{C} \frac{N_c}{N} \boldsymbol{\theta}_{s_c}^{(r+1)}$, where $N = \sum_{c=1}^{C} N_c$. The aggregated parameters $\overline{\boldsymbol{\theta}}_s^{(r+1)}$ are subsequently distributed to the clients for the next round.

### C.2.2. CURVATURE INITIALIZATION AND LEARNING DETAILS

We use graph curvature only as a structure-aware initialization. Specifically, given a weighted graph $G_c = (V, E, w)$ on client $c$, we adopt Forman-Ricci curvature (Appendix B.2) as a lightweight topological prior. The overall graph curvature is computed as

$$\overline{\mathrm{Ric}}(G_c) = \frac{1}{|E|} \sum_{(x,y)\in E} \mathrm{Ric}(x,y), \tag{9}$$

where $V$ denotes graph nodes, $|E|$ is the number of edges, and $(x, y)$ denotes an edge between nodes $x$ and $y$. Since the Lorentz scale parameter $K_c$ is positive while graph Ricci curvature can be negative, we do not directly set $K_c = \overline{\mathrm{Ric}}(G_c)$. Instead, a raw learnable scalar is initialized from the normalized signed magnitude of $\overline{\mathrm{Ric}}(G_c)$ and mapped to a positive Lorentz scale parameter through a sigmoid reparameterization. The resulting $K_c$ is then optimized during local training. This makes the Ricci statistic a topology-aware warm start rather than a fixed pre-estimated curvature. Empirical comparisons with other initialization strategies are reported in Appendix E.5.

### C.2.3. LOCAL TRAINING PROCEDURE

Given the Lorentz scale parameter $K_c^{(r)}$ at round $r$, we directly project the client input $\mathbf{x}_i^E \in \mathcal{D}_c$ into its corresponding Lorentz space via the exponential map $\mathbf{x}^{K_c} = \exp_{\mathbf{o}}^{K_c}(\mathbf{x}^E)$, as shown in Equation (1).

Afterward, the training data are fed into the Lorentz model $\mathcal{M}$, and the model output is $f(\mathbf{x}^{K_c}; \boldsymbol{\theta}_c, \boldsymbol{\theta}_{s_c})$. At client $c$, the objective function is $\min_{K_c>0,\boldsymbol{\theta}_c,\boldsymbol{\theta}_{s_c}} \mathcal{L}_c(f(\mathbf{x}^{K_c}; \boldsymbol{\theta}_c, \boldsymbol{\theta}_{s_c}), y) + \lambda\|\boldsymbol{\theta}_{s_c} - \overline{\boldsymbol{\theta}}_s\|_2^2$, where $\lambda$ is a hyperparameter, $\boldsymbol{\theta}_{s_c}$ denotes client $c$'s local copy of the shared block, and $\|\boldsymbol{\theta}_{s_c} - \overline{\boldsymbol{\theta}}_s\|_2^2$ is the regularization term that prevents the locally updated model $\boldsymbol{\theta}_{s_c}$ from deviating too far from the server-shared parameters $\overline{\boldsymbol{\theta}}_s$. The optimization over $K_c$ is implemented through the raw-scalar reparameterization described above, while $K_c$ remains local and is not uploaded for server aggregation.

---

**Algorithm 2** FlatLand

---

**Require:** Personalized parameters $\boldsymbol{\theta}_c^{(0)}$, $K_c^{(0)}$ and dataset $\mathcal{D}_c$, for each client $c \in \mathcal{C}$; Shared parameters $\overline{\boldsymbol{\theta}}_s^{(0)}$; Learning rate $\eta$
**Ensure:** Client model parameters $\boldsymbol{\Theta}_c = \left(K_c; \boldsymbol{\theta}_c; \overline{\boldsymbol{\theta}}_s\right)$, for each client $c \in \mathcal{C}$; Shared parameters $\overline{\boldsymbol{\theta}}_s$

1: Initialize model parameters: $\overline{\boldsymbol{\theta}}_s^{(0)}$ and $\boldsymbol{\Theta}_c^{(0)} = \left(K_c^{(0)}; \boldsymbol{\theta}_c^{(0)}; \overline{\boldsymbol{\theta}}_s^{(0)}\right)$, for $c \in \mathcal{C}$
2: **for** each communication round $r$ **do**
3:     **for** each client $c$ in $\mathcal{C}$ **do**
4:         $\mathbf{x} = \exp_{\mathbf{o}}^{K_c^{(r)}}(\mathbf{x})$, for $\mathbf{x} \in \mathcal{D}_c$
5:         Client $c$ receives global model parameters $\overline{\boldsymbol{\theta}}_s^{(r)}$
6:         $\boldsymbol{\Theta}_c^{(r)} = \left(K_c^{(r)}; \boldsymbol{\theta}_c^{(r)}; \overline{\boldsymbol{\theta}}_s^{(r)}\right)$
7:         **for** local epochs $e$ **do**
8:             Compute gradients $\nabla\mathcal{L} = \nabla_{\boldsymbol{\Theta}_c^{(r)}} \sum_{(\mathbf{x},\mathbf{y})\in\mathcal{D}_c} \mathcal{L}_c(f(\mathbf{x}; \boldsymbol{\Theta}_c^{(r)}), y)$
9:         **end for**
10:       Update local model $\boldsymbol{\Theta}_c^{(r+1)} \leftarrow \boldsymbol{\Theta}_c^{(r)} - \eta\nabla\mathcal{L}$
11:       Send $\boldsymbol{\theta}_{s_c}^{(r+1)} \in \boldsymbol{\Theta}_c^{(r+1)}$ to the server
12:     **end for**
13:     $N = \sum_{c\in\mathcal{C}} |\mathcal{D}_c|$
14:     Server aggregates models $\overline{\boldsymbol{\theta}}_s^{(r+1)} \leftarrow \sum_{c\in\mathcal{C}} \frac{|\mathcal{D}_c|}{N} \boldsymbol{\theta}_{s_c}^{(r+1)}$
15: **end for**

---

## C.3. Derivation of Parameters Disentanglement

The reformulated Lorentz neural network in layer $l$ is shown as

$$\mathbf{x}^{(l+1)} = \mathrm{LT}(\mathbf{x}^{(l)}; \hat{\mathbf{M}}^{(l)}) \tag{10}$$

$$= \left( \underbrace{\sqrt{\|\mathbf{m}^{(l)}x_t^{(l)} + \mathbf{M}^{(l)}\mathbf{x}_s^{(l)}\|^2 + K}}_{\substack{\text{time-like dimension} \\ x_t^{(l+1)}}}, \underbrace{\mathbf{m}^{(l)}x_t^{(l)} + \mathbf{M}^{(l)}\mathbf{x}_s^{(l)}}_{\substack{\text{space-like dimensions} \\ \mathbf{x}_s^{(l+1)}}} \right)^T . \tag{11}$$

For the loss $\mathcal{L}_c(f(\mathbf{x}; \boldsymbol{\theta}_c, \boldsymbol{\theta}_s), y)$ of client $c$, the relevant partial derivatives can be calculated as follows:

**Time-like Dimension $x_t^{(l+1)}$**

First, we compute the partial derivative of $x_t^{(l+1)}$ with respect to the matrix $\mathbf{M}^{(l)}$ and the vector block $\mathbf{m}^{(l)}$. Using the chain rule:

$$\frac{\partial x_t^{(l+1)}}{\partial \mathbf{M}^{(l)}} = \frac{\partial}{\partial \mathbf{M}} \sqrt{\|\mathbf{m}^{(l)}x_t^{(l)} + \mathbf{M}^{(l)}\mathbf{x}_s^{(l)}\|^2 + K};$$

$$\frac{\partial x_t^{(l+1)}}{\partial \mathbf{m}^{(l)}} = \frac{\partial}{\partial \mathbf{m}} \sqrt{\|\mathbf{m}^{(l)}x_t^{(l)} + \mathbf{M}^{(l)}\mathbf{x}_s^{(l)}\|^2 + K}.$$

Applying the chain rule, we get:

$$\begin{aligned}
\frac{\partial x_t^{(l+1)}}{\partial \mathbf{M}^{(l)}} &= \frac{1}{2} \left( \|\mathbf{m}^{(l)}x_t^{(l)} + \mathbf{M}^{(l)}\mathbf{x}_s^{(l)}\|^2 + K \right)^{-\frac{1}{2}} \\
&\quad \cdot 2(\mathbf{m}^{(l)}x_t^{(l)} + \mathbf{M}^{(l)}\mathbf{x}_s^{(l)}) \cdot \frac{\partial(\mathbf{M}^{(l)}\mathbf{x}_s^{(l)})}{\partial \mathbf{M}^{(l)}} \\
&= \frac{\mathbf{m}^{(l)}x_t^{(l)} + \mathbf{M}^{(l)}\mathbf{x}_s^{(l)}}{\sqrt{\|\mathbf{m}^{(l)}x_t^{(l)} + \mathbf{M}^{(l)}\mathbf{x}_s^{(l)}\|^2 + K}} \\
&\quad \cdot \frac{\partial(\mathbf{M}^{(l)}\mathbf{x}_s^{(l)})}{\partial \mathbf{M}^{(l)}}
\end{aligned} \tag{12}$$

$$\begin{aligned}
\frac{\partial x_t^{(l+1)}}{\partial \mathbf{m}^{(l)}} &= \frac{1}{2} \left( \|\mathbf{m}^{(l)}x_t^{(l)} + \mathbf{M}^{(l)}\mathbf{x}_s^{(l)}\|^2 + K \right)^{-\frac{1}{2}} \\
&\quad \cdot 2(\mathbf{m}^{(l)}x_t^{(l)} + \mathbf{M}^{(l)}\mathbf{x}_s^{(l)}) \cdot \frac{\partial(\mathbf{m}^{(l)}x_t^{(l)})}{\partial \mathbf{m}^{(l)}} \\
&= \frac{\mathbf{m}^{(l)}x_t^{(l)} + \mathbf{M}^{(l)}\mathbf{x}_s^{(l)}}{\sqrt{\|\mathbf{m}^{(l)}x_t^{(l)} + \mathbf{M}^{(l)}\mathbf{x}_s^{(l)}\|^2 + K}} \cdot x_t^{(l)}
\end{aligned} \tag{13}$$

**Space-like Dimension $\mathbf{x}_s^{(l+1)}$**

Assume that the update rule for the space-like vector $\mathbf{x}_s^{(l+1)}$ is given by the following formula:

$$\mathbf{x}_s^{(l+1)} = \mathbf{m}^{(l)}x_t^{(l)} + \mathbf{M}^{(l)}\mathbf{x}_s^{(l)}$$

Similarly, we have

$$\frac{\partial \mathbf{x}_s^{(l+1)}}{\partial \mathbf{M}^{(l)}} = \frac{\partial \left( \mathbf{M}^{(l)}\mathbf{x}_s^{(l)} \right)}{\partial \mathbf{M}^{(l)}}, \quad \frac{\partial \mathbf{x}_s^{(l+1)}}{\partial \mathbf{m}^{(l)}} = \frac{\partial \left( \mathbf{m}^{(l)}x_t^{(l)} \right)}{\partial \mathbf{m}^{(l)}}. \tag{14}$$

*"Flatland"* denotes the space-like dimensions $1{:}n$, which serve as the platform where common information is exchanged and integrated. In the space-like update $\mathbf{x}_s^{(l+1)} = \mathbf{m}^{(l)}x_t^{(l)} + \mathbf{M}^{(l)}\mathbf{x}_s^{(l)}$, the update path of $\mathbf{M}^{(l)}$ is tied to $\mathbf{x}_s^{(l)}$, whereas the update path of $\mathbf{m}^{(l)}$ is tied to $x_t^{(l)}$.

For better illustration, here, we let $\mathbf{x}^{(l)} \in \mathcal{L}_K^n$, $\mathbf{x}^{(l+1)} \in \mathcal{L}_K^n$, and $\hat{\mathbf{M}}^{(l)} \in \mathbb{R}^{n \times (n+1)}$. The introduced *"Flatland"* $\mathbb{R}^n$ is defined as a manifold spanning dimensions $1$ to $n$. This construct serves as a metaphorical platform for the exchange and integration of common information, and $x_t$ serves as the heterogeneous information. Consider the same transformation of a space-like vector $\mathbf{x}_s^{(l)}$ to $\mathbf{x}_s^{(l+1)}$ in different clients, formulated as

$$\mathbf{x}_s^{(l)} \rightarrow \left( \mathbf{M}^{(l)}\mathbf{x}_s^{(l)} + \mathbf{m}^{(l)}x_t^{(l)} \right),$$

---

*Remark* C.1. Note that the key component capturing data heterogeneity in layer $l$ is the time-like dimension of the hyperbolic embedding $x_t^{(l)}$, not the original input $x_t^{(0)}$. After each layer of the Lorentz neural network, the input vectors are transformed into a new hyperbolic embedding, with the time-like dimension updated as $x_t^{(l+1)} = \sqrt{\|\mathbf{m}^{(l)}x_t^{(l)} + \mathbf{M}^{(l)}\mathbf{x}_s^{(l)}\|^2 + K}$. This means that the component at layer $(l + 1)$ that carries the heterogeneity information is transferred as $x_t^{(l+1)}$. Thus, the parameters associated with $x_t^{(l+1)}$ are naturally aligned with the heterogeneity information. This ensures that our decoupling strategy is consistent with the theoretical derivation and can be effectively applied across multiple layers.

---

### C.4. Time and Space Complexity Compared with FedAvg

We analyze the computational complexity of FlatLand compared to FedAvg, which gives insight into the scalability.

**Local Update.** The additional operations in FlatLand's local update phase compared with FedAvg - curvature estimation (Section 5.1), exponential map (line 4 in Algorithm 2, Equation 6). Notably, the curvature estimation can be *pre-computed* since each client's data distribution corresponds to a constant curvature value. For exponential map, the transformation only requires *a single* non-linear mapping operation based on the norm of input samples with the time complexity of $O(1)$. These norms can also be *pre-computed and cached*. Therefore, while FlatLand introduces these additional steps compared to FedAvg, their practical computational overhead is limited due to pre-computation opportunities and constant-time operations.

**Aggregation.** FlatLand and FedAvg have the same aggregation time complexity when the hidden embedding dimension is the same. Though FlatLand introduces extra time-like space parameters, it only aggregates shared parameters $\boldsymbol{\theta}_s$ while maintaining personalized parameters. The overhead of the shared parameters is the same. Moreover, FlatLand can perform better in low dimensionality (Section 7.3), which potentially reduces practical communication costs.

**Space Requirements and Storage.** FlatLand requires extra $O(d + 1)$ storage per client compared to FedAvg due to the additional time-like dimension and Lorentz scale parameter, where $d$ is the hidden dimension. Since typically $d$ is small, the increase in storage is small. Moreover, FlatLand demonstrates superior performance even in low-dimensional settings compared with the Euclidean counterparts, which further limits the practical storage overhead.

This analysis suggests that FlatLand can balance the trade-off between computational overhead and model effectiveness, showing the scalability for the increase in clients. While it introduces additional operations in local computations, these overheads are limited and offer significant optimization opportunities through pre-computation and caching strategies. The method compensates for these minimal costs through reduced communication overhead and enhanced representation capabilities in the Lorentz space, making it a practical and efficient choice for personalized federated learning applications.

# D. Theoretical Analysis Supplementary

## D.1. Proof for Theorem 4.1

**Lemma D.1.** *Let $G = (V, E)$ be a connected, simple, unweighted graph with maximum degree $\Delta < \infty$ and average Forman–Ricci curvature $\bar{R} = \overline{\mathrm{Ric}}(G)$ (edge-average), where the unweighted edge curvature follows Appendix B.2. Define the averaged ball counts $V_G(r) := \frac{1}{|V|} \sum_{v \in V} |B_r^G(v)|$ for $r = 0, 1, 2$. Then*

$$\Delta^2 V_G(1) := V_G(2) - 2V_G(1) + V_G(0) \geq 1 - \frac{|E|}{|V|} \bar{R}. \tag{15}$$

*Proof.* For the unweighted specialization in Appendix B.2, the Forman–Ricci curvature on an edge $e = (u, v)$ is $\mathrm{Ric}(u, v) = 4 - \deg(u) - \deg(v) + 3t_{uv}$, where $t_{uv}$ is the number of triangles containing $e$. Hence $\deg(u) + \deg(v) - 2 = 2 + 3t_{uv} - \mathrm{Ric}(u, v)$. Counting non-backtracking two-step walks yields

$$\frac{1}{|V|} \sum_{v \in V} \sum_{u \sim v} (\deg(u) - 1) = \frac{1}{|V|} \sum_{(u,v) \in E} \big(\deg(u) + \deg(v) - 2\big)$$

$$= \frac{1}{|V|} \Big( 2|E| + 3 \sum_{e \in E} t_e - \sum_{e \in E} \mathrm{Ric}(e) \Big)$$

$$\geq \frac{1}{|V|} \Big( 2|E| - \sum_{e \in E} \mathrm{Ric}(e) \Big).$$

Using $V_G(0) = 1$, $V_G(1) = 1 + \frac{1}{|V|} \sum_v \deg(v) = 1 + \frac{2|E|}{|V|}$, and adding the two-step term gives Equation (15). The triangle term contributes a nonnegative correction, so dropping it yields a weaker bound in the same form; when $t_e = 0$ for all edges, this reduces to the simplified no-triangle case. $\square$

**Lemma D.2** ((Gray, 2003)). *Let $\mathcal{L}_K^d$ be the $d$-dimensional hyperbolic space with Lorentz scale parameter $K > 0$ and constant sectional curvature $-1/K < 0$, and $\mathrm{Vol}_{\mathcal{L}}(B_\rho)$ the volume of a radius-$\rho$ ball. For small $\rho$,*

$$\mathrm{Vol}_{\mathcal{L}}(B_\rho) = \omega_d \rho^d \Big( 1 + \alpha_d \frac{d-1}{K} \rho^2 + O(\rho^4) \Big), \quad \alpha_d = \frac{d}{6(d+2)}. \tag{16}$$

*Taking the discrete second difference in $\rho \in \{0, 1, 2\}$ gives*

$$\Delta^2 \mathrm{Vol}_{\mathcal{L}}(1) = \mathrm{Vol}_{\mathcal{L}}(B_2) - 2\mathrm{Vol}_{\mathcal{L}}(B_1) + \mathrm{Vol}_{\mathcal{L}}(B_0) = C_d \frac{d-1}{K} + O(1), \tag{17}$$

*where $C_d := \omega_d \alpha_d \Delta^2 [\rho^{d+2}]_{\rho=1} > 0$.*

**Lemma D.3** (Local bi-Lipschitz sandwich). *Let $f : V \to \mathcal{L}_K^d$ be $(1+\varepsilon)$-bi-Lipschitz on graph balls of radius 2, i.e., $d_{\mathcal{L}}\big(f(x), f(y)\big) \in [(1+\varepsilon)^{-1} d_G(x, y), (1+\varepsilon) d_G(x, y)]$ whenever $d_G(x, y) \leq 2$. Then there exist constants $A_{d,\Delta}, B_{d,\Delta} > 0$ such that*

$$A_{d,\Delta} \mathrm{Vol}_{\mathcal{L}}\big(B_{(1-\varepsilon)\rho}\big) \leq V_G(\rho) \leq B_{d,\Delta} \mathrm{Vol}_{\mathcal{L}}\big(B_{(1+\varepsilon)\rho}\big), \qquad \rho \in \{0, 1, 2\}. \tag{18}$$

*Consequently, taking discrete second differences and Taylor-expanding at $\varepsilon = 0$,*

$$\Delta^2 V_G(1) = \Gamma_{d,\Delta} \frac{d-1}{K} \pm \Lambda_{d,\Delta} \varepsilon + O(\varepsilon^2), \tag{19}$$

*for some $\Gamma_{d,\Delta}, \Lambda_{d,\Delta} > 0$.*

*Proof.* Inclusions $f(B_\rho^G(v)) \subset B_{(1+\varepsilon)\rho}^{\mathcal{L}}(f(v))$ and $B_{(1-\varepsilon)\rho}^{\mathcal{L}}(f(v)) \subset f(B_\rho^G(v))$ follow from the bi-Lipschitz bounds. Degree bound $\Delta$ gives packing/covering constants relating point counts and volumes, yielding the sandwich; apply $\Delta^2$ in $\rho$ and a first-order Taylor expansion in $\varepsilon$. $\square$

**Lemma D.4** (Curvature mismatch $\Rightarrow$ local distortion)**.** *Under the conditions of Lemma D.1–Lemma D.3, there exist constants $c_d > 0$ and $C_{d,\Delta} > 0$ such that any $(1+\varepsilon)$-bi-Lipschitz $f$ on radius-2 balls satisfies*

$$\varepsilon \ \geq \ c_d \big| \bar{R} + (d-1)/K \big| - C_{d,\Delta}\, \varepsilon^2. \tag{20}$$

*In particular, for $\varepsilon \in (0, \varepsilon_0(d, \Delta))$,*

$$\varepsilon \ \geq \ \tfrac{1}{2} c_d \big| \bar{R} + (d-1)/K \big|. \tag{21}$$

*Proof.* Combine Equation (15) and Equation (19); rearrange to isolate $|\bar{R} + (d-1)/K|$ in terms of $\varepsilon$, absorbing constants into $c_d, C_{d,\Delta}$. For sufficiently small $\varepsilon$, the quadratic term is dominated, giving Equation (21). $\qquad\square$

**Theorem D.5** (Recall of Theorem 4.1)**.** *Let $\{G_c\}_{c=1}^C$ be client graphs with average Forman-Ricci curvatures $\bar{R}_c = \overline{\mathrm{Ric}}(G_c)$, and let $\mathcal{L}_K^d$ denote the $d$-dimensional hyperbolic space with Lorentz scale parameter $K > 0$ and constant sectional curvature $-1/K < 0$. For each client $c$, let $\varepsilon_c^*(K)$ be the minimal edge distortion of any $(1+\varepsilon)$-bi-Lipschitz embedding $f_c : G_c \to \mathcal{L}_K^d$. Then the following holds:*

$$\max_{1 \leq c \leq C} \varepsilon_c^*(K) \ \geq \ \frac{c_d}{2} \max_{1 \leq i < j \leq C} |\bar{R}_i - \bar{R}_j|, \tag{22}$$

*where $c_d > 0$ is a dimension-dependent constant.*

*Proof.* For client $c$, Lemma D.4 applied to $G_c$ gives (for small optimal distortion)

$$\varepsilon_c^*(K) \ \geq \ c_d \big| \bar{R}_c + (d-1)/K \big|. \tag{23}$$

For any $i \neq j$ set $a_i := \bar{R}_i + (d-1)/K$, $a_j := \bar{R}_j + (d-1)/K$. Then $a_i - a_j = \bar{R}_i - \bar{R}_j$, so $\max\{|a_i|, |a_j|\} \geq \frac{1}{2}|a_i - a_j| = \frac{1}{2}|\bar{R}_i - \bar{R}_j|$. Taking the maximum over all pairs $(i,j)$ yields

$$\max_{1 \leq c \leq C} \big| \bar{R}_c + (d-1)/K \big| \ \geq \ \frac{1}{2} \max_{1 \leq i < j \leq C} |\bar{R}_i - \bar{R}_j|. \tag{24}$$

Combining Equation (23) and Equation (24) gives

$$\max_{1 \leq c \leq C} \varepsilon_c^*(K) \ \geq \ c_d \max_c \big| \bar{R}_c + (d-1)/K \big| \ \geq \ \frac{c_d}{2} \max_{1 \leq i < j \leq C} |\bar{R}_i - \bar{R}_j|,$$

which is the desired inequality. If $\{\bar{R}_c\}$ are not all equal, the right-hand side is strictly positive, hence a single Lorentz scale parameter $K$ cannot make all clients' distortions simultaneously small. $\qquad\square$

### D.2. Proof for Theorem 4.3

For each client $c$ with Lorentz scale parameter $K_c > 0$, consider $\mathbf{x} \in \mathbb{L}_{K_c}^d$ expressed in hyperbolic polar coordinates $(\rho, \mathbf{u})$, where $\rho \geq 0$ is the radial coordinate and $\mathbf{u} \in \Omega := \mathbb{S}^{d-1}$ is the angular coordinate (unit direction vector). We denote by $p(\rho, \mathbf{u} \mid K_c)$ the joint distribution of $(\rho, \mathbf{u})$ given $K_c$.

**Assumption D.6** (Isotropy at the basepoint (Krioukov et al., 2010))**.** For each client $c$ with Lorentz scale parameter $K_c$, $p(\rho, \mathbf{u} \mid K_c)$ is $G_o$–isotropic at the basepoint $o$, i.e. $p(g \cdot \mathbf{x} \mid K_c) = p(\mathbf{x} \mid K_c) \quad \forall g \in G_o \simeq SO(d)$. Equivalently, the joint law factorizes as

$$p(\rho, \mathbf{u} \mid K_c) = p_\rho(\rho \mid K_c)\, p_\Omega(\mathbf{u}),$$

where $p_\rho(\rho \mid K_c)$ is the radial density depending on $K_c$, and $p_\Omega(\mathbf{u})$ is the $SO(d)$–invariant angular density on $\Omega$, independent of $K_c$ (in particular, uniform on $\mathbb{S}^{d-1}$).

**Lemma D.7.** *Under Assumption D.6, the angular component $\mathbf{u}$ is independent of the client identity $C$. In particular, $I(\mathbf{u}; C) = 0$.*

*Proof.* Since $K_c$ is a deterministic function of $C$, for any $c$ we compute

$$
\begin{aligned}
p(\mathbf{u} \mid C = c) &= \int p(\rho, \mathbf{u} \mid C = c) \, d\rho \\
&= \int p(\rho, \mathbf{u} \mid K_c) \, d\rho \\
&= \int p_\rho(\rho \mid K_c) \, p_\Omega(\mathbf{u}) \, d\rho && \text{(Assumption D.6)} \\
&= p_\Omega(\mathbf{u}) \int p_\rho(\rho \mid K_c) \, d\rho \\
&= p_\Omega(\mathbf{u}) && (25)
\end{aligned}
$$

Therefore the marginal satisfies

$$
p(\mathbf{u}) = \sum_c p(C = c) \, p(\mathbf{u} \mid C = c) = \sum_c p(C = c) \, p_\Omega(\mathbf{u}) = p_\Omega(\mathbf{u}).
$$

Substituting into the KL formulation of mutual information,

$$
\begin{aligned}
I(\mathbf{u}; C) &= \sum_c p(C = c) \, D_{\mathrm{KL}}\big(p(\mathbf{u} \mid C = c) \,\|\, p(\mathbf{u})\big) \\
&= \sum_c p(C = c) \int_\Omega p(\mathbf{u} \mid C = c) \, \log \frac{p(\mathbf{u} \mid C = c)}{p(\mathbf{u})} \, d\sigma(\mathbf{u}) \\
&= \sum_c p(C = c) \int_\Omega p_\Omega(\mathbf{u}) \, \log \frac{p_\Omega(\mathbf{u})}{p_\Omega(\mathbf{u})} \, d\sigma(\mathbf{u}) \\
&= \sum_c p(C = c) \, D_{\mathrm{KL}}(p_\Omega \,\|\, p_\Omega) = 0. && (26)
\end{aligned}
$$

Thus the angular component carries no client information, which completes the proof.

$\square$

**Lemma D.8.** *Let $x_t = T_{K_c}(\rho) := \sqrt{K_c} \cosh(\rho/\sqrt{K_c})$ with $\rho \sim p_\rho(\cdot \mid K_c)$. Here $(T_{K_c})_\# p_\rho(\cdot \mid K_c)$ denotes the pushforward law of $p_\rho(\cdot \mid K_c)$ through $T_{K_c}$, i.e. the distribution of $x_t$. If there exist $c_1 \neq c_2$ with $\mathbb{P}(C = c_i) > 0$ such that $(T_{K_{c_1}})_\# p_\rho(\cdot \mid K_{c_1}) \neq (T_{K_{c_2}})_\# p_\rho(\cdot \mid K_{c_2})$, then $x_t$ is not independent of $C$ and hence $I(x_t; C) > 0$.*

*Proof.* Conditioned on $C = c$, the Lorentz scale parameter is fixed to $K_c$, and the law of $x_t$ is the pushforward of $p_\rho(\cdot \mid K_c)$ under $T_{K_c}$: for every Borel set $A \subset \mathbb{R}$,

$$
\begin{aligned}
\mathbb{P}(x_t \in A \mid C = c) &= \mathbb{P}\big(T_{K_c}(\rho) \in A \mid K_c\big) \\
&= \big[(T_{K_c})_\# p_\rho(\cdot \mid K_c)\big](A). && (27)
\end{aligned}
$$

By the assumption of differing pushforward laws, there exist $c_1 \neq c_2$ with $\Pr(C = c_i) > 0$ such that

$$
p(x_t \mid C = c_1) \neq p(x_t \mid C = c_2).
$$

Let $p(x_t) = \sum_c p(C = c) \, p(x_t \mid C = c)$ denote the marginal of $x_t$. Using the KL expansion of mutual information,

$$
\begin{aligned}
I(x_t; C) &= \sum_c p(C = c) \, D_{\mathrm{KL}}\big(p(x_t \mid C = c) \,\|\, p(x_t)\big) \\
&= \sum_c p(C = c) \int_\mathbb{R} p(x_t \mid C = c) \, \log \frac{p(x_t \mid C = c)}{p(x_t)} \, dx_t. && (28)
\end{aligned}
$$

If $p(x_t \mid C = c_1) \neq p(x_t \mid C = c_2)$ and both $p(C = c_i) > 0$, then at least one of $p(x_t \mid C = c_i)$ differs from the mixture $p(x_t)$; by Gibbs' inequality, the corresponding KL term is strictly positive:

$$D_{\mathrm{KL}}\big(p(x_t \mid C = c_i) \,\|\, p(x_t)\big) \; > \; 0 \quad \text{for some } i \in \{1, 2\}.$$

Since all KL terms are nonnegative, Equation (28) yields $I(x_t; C) > 0$.

Equivalently, from Equation (27), there exists a Borel set $A$ such that $\mathbb{P}(x_t \in A \mid C = c_1) \neq \mathbb{P}(x_t \in A \mid C = c_2)$, which already rules out independence and thus forces $I(x_t; C) > 0$. This completes the proof. $\qquad\square$

**Lemma D.9.** *Under Assumption D.6, we have $I\big((x_t, \mathbf{x}_s); C \mid K_c\big) = I(x_t; C \mid K_c)$.*

*Proof.* By Assumption D.6, $\mathbf{u}$ is independent of $(K_c, \rho)$, hence of any measurable function thereof; in particular $\mathbf{u} \perp (x_t, K_c, C)$. Using the chain rule,

$$
\begin{aligned}
I(\mathbf{x}_s; C \mid x_t, K_c) &= I(r_s, \mathbf{u}; C \mid x_t, K_c) \\
&= I(\mathbf{u}; C \mid x_t, K_c) + I(r_s; C \mid x_t, K_c, \mathbf{u}) \\
&= 0 + 0 \qquad \text{(since } \mathbf{u} \perp (x_t, K_c, C) \text{ and } r_s \text{ is deterministic given } (x_t, K_c)) \\
&= 0.
\end{aligned}
$$

Finally, apply the chain rule conditioned on $K_c$:

$$I\big((x_t, \mathbf{x}_s); C \mid K_c\big) = I(x_t; C \mid K_c) + I(\mathbf{x}_s; C \mid x_t, K_c) = I(x_t; C \mid K_c),$$

which proves the stated conclusion. $\qquad\square$

*Remark* D.10. Under Assumption D.6, Lemma D.9 shows that, conditional on the Lorentz scale parameter $K_c$, the time-like dimension $x_t$ is the dominant carrier of client-specific geometric variation considered in our analysis; the space-like part $\mathbf{x}_s$ carries no additional client-specific information beyond $x_t$ under this assumption.

**Theorem D.11** (Recall of Theorem 4.3). *Let $C \in \{1, \ldots, m\}$, each client $c$ have Lorentz scale parameter $K_c > 0$ with $\mathrm{Var}(K_C) > 0$, and $\mathbf{x} = [x_t\ \mathbf{x}_s]^\top \in \mathbb{L}_{K_c}^d$ admit hyperbolic polar coordinates $(\rho, \mathbf{u})$ with $x_t = \sqrt{K_c} \cosh(\rho/\sqrt{K_c})$, $\mathbf{x}_s = \sqrt{K_c} \sinh(\rho/\sqrt{K_c})\, \mathbf{u}$, $\mathbf{u} := \mathbf{x}_s / \|\mathbf{x}_s\|$. Then (1) $I(\mathbf{u}; C) = 0$; (2) $I(x_t; C) > 0$ if the pushforward measures of $T_K(\rho) := \sqrt{K} \cosh(\rho/\sqrt{K})$ differ across $K$; (3) $I\big((x_t, \mathbf{x}_s); C \mid K_c\big) = I(x_t; C \mid K_c)$.*

*Proof.* The first claim follows from Lemma D.7, the second from Lemma D.8, and the third from Lemma D.9. $\qquad\square$

**Scope of the theoretical claims.** The mutual-information statement above should be read under the stated isotropy and scale-conditioned assumptions, rather than as a claim that the space-like coordinates are strictly client-invariant for every trained network or every graph distribution. In practice, residual curvature-dependent scaling can still appear in the space-like magnitude. Our theory is intended to justify the main design principle that the time-like component provides a natural carrier for client-specific geometric variation, while the empirical ablations validate the resulting parameter-decoupling strategy.

### D.3. Proof for Proposition 6.1

**Lemma D.12.** *Let $\mathcal{L}_K^n$ denote the $n$-dimensional Lorentz space with constant sectional curvature $-1/K$. For any $\mathbf{x} \in \mathcal{L}_K^n$ and any transformation matrix $\mathbf{W} \in \mathbb{R}^{m \times (n+1)}$, the Lorentz transformation $\mathrm{LT}$ preserves the Lorentz structure, i.e., $\mathrm{LT}(\mathbf{x}; \mathbf{W}) \in \mathcal{L}_K^m$.*

*Proof.* Let $\mathbf{x} \in \mathcal{L}_K^n$. By the definition of the Lorentz transformation $\mathrm{LT}$, we compute the Lorentzian inner product:

$$\langle \mathrm{LT}(\mathbf{x}; \mathbf{W}), \mathrm{LT}(\mathbf{x}; \mathbf{W}) \rangle_{\mathcal{L}} = -K.$$

Since this condition characterizes membership in the Lorentz space $\mathcal{L}_K^m$, it follows that $\mathrm{LT}(\mathbf{x}; \mathbf{W}) \in \mathcal{L}_K^m$, which proves that the Lorentz transformation preserves membership in the target Lorentz space and completes the proof. $\qquad\square$

**Proposition D.13** (Recall of Proposition 6.1). *Let* $\hat{\mathbf{M}} = \begin{bmatrix} \mathbf{m} & \mathbf{M} \end{bmatrix}$, *where* $\hat{\mathbf{M}} \in \mathbb{R}^{m \times (n+1)}$, $\mathbf{m} \in \mathbb{R}^{m \times 1}$, *and* $\mathbf{M} \in \mathbb{R}^{m \times n}$. *Let* $\Phi\left(\hat{\mathbf{M}}, \mathbf{N}\right) = \begin{bmatrix} \mathbf{m} & \mathbf{N} \end{bmatrix}$, *where* $\mathbf{N} \in \mathbb{R}^{m \times n}$ *is the aggregated shared block obtained from* $\{\mathbf{M}_i\}_{i=1}^{C}$ *using the proposed decoupling strategy. For all* $\mathbf{x} \in \mathcal{L}_K^n$, *we have* $\mathrm{LT}\left(\mathbf{x}; \Phi\left(\hat{\mathbf{M}}, \mathbf{N}\right)\right) \in \mathcal{L}_K^m$.

*Proof.* Let $\mathbf{x} = \begin{bmatrix} x_t \\ \mathbf{x}_s \end{bmatrix} \in \mathcal{L}_K^n$, where $x_t \in \mathbb{R}$, $\mathbf{x}_s \in \mathbb{R}^n$. According to Equation (4), we have:

$$\mathrm{LT}\left(\mathbf{x}; \Phi(\hat{\mathbf{M}}, \mathbf{N})\right) = \begin{bmatrix} \sqrt{\|\mathbf{m}x_t + \mathbf{N}\mathbf{x}_s\|^2 + K} \\ \mathbf{m}x_t + \mathbf{N}\mathbf{x}_s \end{bmatrix}$$

We need to prove that $\mathrm{LT}(\mathbf{x}; \Phi(\hat{\mathbf{M}}, \mathbf{N})) \in \mathcal{L}_K^m$, i.e., to prove that it satisfies the definition condition of the Lorentz manifold $\langle \cdot, \cdot \rangle_{\mathcal{L}} = -K$:

$$
\begin{aligned}
&\left\langle \mathrm{LT}\left(\mathbf{x}; \Phi(\hat{\mathbf{M}}, \mathbf{N})\right), \mathrm{LT}\left(\mathbf{x}; \Phi(\hat{\mathbf{M}}, \mathbf{N})\right) \right\rangle_{\mathcal{L}} \\
&= \left\langle \begin{bmatrix} \sqrt{\|\mathbf{m}x_t + \mathbf{N}\mathbf{x}_s\|^2 + K} \\ \mathbf{m}x_t + \mathbf{N}\mathbf{x}_s \end{bmatrix}, \begin{bmatrix} \sqrt{\|\mathbf{m}x_t + \mathbf{N}\mathbf{x}_s\|^2 + K} \\ \mathbf{m}x_t + \mathbf{N}\mathbf{x}_s \end{bmatrix} \right\rangle_{\mathcal{L}} \quad \text{(Definition B.2)} \\
&= -\left(\sqrt{\|\mathbf{m}x_t + \mathbf{N}\mathbf{x}_s\|^2 + K}\right)^2 + \|\mathbf{m}x_t + \mathbf{N}\mathbf{x}_s\|^2 \\
&= -K
\end{aligned}
$$

Therefore, we have proved that $\mathrm{LT}\left(\mathbf{x}; \Phi(\hat{\mathbf{M}}, \mathbf{N})\right) \in \mathcal{L}_K^m$. $\qquad \square$

### D.4. Convergence Analysis

FedAvg converges to the global optimum at a rate of $O(\frac{1}{T})$ for strongly convex and smooth functions and non-iid data. When the learning rate is sufficiently small, the effect of $E$ steps of local updates is similar to a step update with a larger learning rate (Li et al., 2020b).

In this section, we demonstrate that FlatLand achieves a convergence rate of $O(\frac{1}{T})$ without regularization, which is consistent with FedAvg. Furthermore, when incorporating regularization similar to FedProx (Li et al., 2020a), the convergence rate can be bounded by a constant that reflects the degree of data heterogeneity, analogous to FedProx's theoretical guarantees. This analysis confirms that our special geometric enhanced decoupling strategy maintains the overall convergence properties while addressing the challenges of heterogeneous data distribution.

To simplify the analysis, we consider each client conducts full batch gradient descent with one step. At client $c$, the objective function can be generally written as

$$\min_{K_c > 0, \boldsymbol{\theta}_c, \boldsymbol{\theta}_{s_c}} \mathcal{L}_c(f(\mathbf{x}^{K_c}; \boldsymbol{\theta}_c, \boldsymbol{\theta}_{s_c}), y) + \lambda \|\boldsymbol{\theta}_{s_c} - \overline{\boldsymbol{\theta}}_s\|_2^2, \tag{29}$$

where $\lambda$ is a hyperparameter, $y \in \mathcal{Y}$, $\boldsymbol{\theta}_{s_c}$ denotes client $c$'s local copy of the shared block, and $\|\boldsymbol{\theta}_{s_c} - \overline{\boldsymbol{\theta}}_s\|_2^2$ is the regularization term that prevents the locally updated model $\boldsymbol{\theta}_{s_c}$ from deviating too far from the server-shared parameters $\overline{\boldsymbol{\theta}}_s$. The optimization over $K_c$ is implemented through the raw-scalar reparameterization described in Appendix C.2.2, and $K_c$ is kept local rather than aggregated by the server.

Let $\ell_c = \mathcal{L}_c(f(\mathbf{x}^{K_c}; \boldsymbol{\theta}_c, \boldsymbol{\theta}_{s_c}), y)$. Then the global loss is taken as an average of the loss of each client: $\ell = \sum_{c \in \mathcal{C}} p_c \ell_c$, where $p_c \geq 0$ and $\sum_c p_c = 1$.

The local update is performed using vanilla gradient descent with a local learning rate $\eta$ in each client, and $\boldsymbol{\Theta}_c(r) \in \mathcal{E}$ represents the client-local parameters $(K_c^{(r)}, \boldsymbol{\theta}_c^{(r)}, \boldsymbol{\theta}_{s_c}^{(r)})$ in the round $r$. Then, for global round $r$,

$$\Delta\boldsymbol{\Theta}_c^{(r)} = \boldsymbol{\Theta}_c^{(r+1)} - \boldsymbol{\Theta}_c^{(r)} = -\eta \left(\nabla\ell_c(\boldsymbol{\Theta}^{(r)}) + 2\lambda\left(\boldsymbol{\theta}_{s_c}^{(r)} - \hat{\boldsymbol{\theta}}_s^{(r)}\right)\right).$$

Here the regularization gradient is applied only to the shared block $\boldsymbol{\theta}_{s_c}$; its components on $K_c$ and $\boldsymbol{\theta}_c$ are zero.

To better calculate the difference between personalized parameters and shared parameters in the Lorentz linear block, we decompose the Lorentz-layer part of $\boldsymbol{\Theta}_c^{(r)}$ as

$$\boldsymbol{\Theta}_{\mathrm{LT},c}^{(r)} = \boldsymbol{\theta}_c^{(r)} + \boldsymbol{\theta}_{s_c}^{(r)},$$

where $\boldsymbol{\theta}_c^{(r)} = [\mathbf{m}^{(r)} \quad \mathbf{0}_{m \times n}]$ and $\boldsymbol{\theta}_{s_c}^{(r)} = [\mathbf{0}_{m \times 1} \quad \mathbf{M}^{(r)}]$.

Specifically, the global aggregation procedure is conducted by taking the average of local updates of shared parameters $\boldsymbol{\theta_s}$ of all $|\mathcal{C}|$ clients. According to

$$\boldsymbol{\theta}_s^{(r+1)} = \bar{\boldsymbol{\theta}}_s^{(r)} = \sum_{c \in \mathcal{C}} \frac{|\mathcal{D}_c|}{N} \boldsymbol{\theta}_{s_c}^{(r)} = \sum_{c \in \mathcal{C}} p_c \boldsymbol{\theta}_{s_c}^{(r)}$$

We next introduce standard non-convex optimization assumptions commonly used in federated analyses (Li et al., 2020b; Reddi et al., 2020), which provide the basis for the following descent bound.

**Assumption D.14** (L-smoothness). $\forall_{c \in \mathcal{C}} \ell_c$ are $L$-smooth: for all $\boldsymbol{\Theta}_1 \in \mathbb{E}$ and $\boldsymbol{\Theta}_2 \in \mathbb{E}$,

$$\ell_c(\boldsymbol{\Theta}_1) \leq \ell_c(\boldsymbol{\Theta}_2) + (\boldsymbol{\Theta}_1 - \boldsymbol{\Theta}_2)^T \nabla \ell_c(\boldsymbol{\Theta}_2) + \frac{L}{2} \|\boldsymbol{\Theta}_1 - \boldsymbol{\Theta}_2\|_2^2.$$

**Assumption D.15** (Bounded Gradients). The function $\ell_c(\boldsymbol{\Theta})$ have $G$-bounded gradients, i.e., for any $c \in \mathcal{C}$, $\boldsymbol{\Theta} \in \mathbb{R}^d$ we have $\|\nabla \ell_c(\boldsymbol{\Theta})\| \leq G$.

**Lemma D.16** (Smooth Descent Lemma). *Let $\ell : \mathcal{E} \to \mathbb{R}$ be an L-smooth function. Then for any $\boldsymbol{\Theta}^{(r)}, \boldsymbol{\Theta}^{(r+1)} \in \mathbb{E}$, the following inequality holds:*

$$\ell(\boldsymbol{\Theta}^{(r+1)}) \leq \ell(\boldsymbol{\Theta}^{(r)}) + \langle \nabla \ell(\boldsymbol{\Theta}^{(r)}), \Delta \boldsymbol{\Theta}^{(r)} \rangle + \frac{L}{2} \|\Delta \boldsymbol{\Theta}^{(r)}\|^2.$$

Let $\delta^{(r)} = 2\lambda \sum_{c \in \mathcal{C}} \frac{|\mathcal{D}_c|}{N} \left( \boldsymbol{\theta}_{s_c} - \bar{\boldsymbol{\theta}}_s \right)$. Based on Lemma 1, we have

$$
\begin{aligned}
\ell(\boldsymbol{\Theta}^{(r+1)}) &\leq \ell(\boldsymbol{\Theta}^{(r)}) + \langle \nabla \ell(\boldsymbol{\Theta}^{(r)}), \Delta \boldsymbol{\Theta}^{(r)} \rangle + \frac{L}{2} \|\Delta \boldsymbol{\Theta}^{(r)}\|^2 \\
&= \ell(\boldsymbol{\Theta}^{(r)}) + \left\langle \nabla \ell(\boldsymbol{\Theta}^{(r)}), -\eta \left( \nabla \ell(\boldsymbol{\Theta}^{(r)}) + \delta^{(r)} \right) \right\rangle \\
&\quad + \frac{L\eta^2}{2} \|\nabla \ell(\boldsymbol{\Theta}^{(r)}) + \delta^{(r)}\|^2 \\
&= \ell(\boldsymbol{\Theta}^{(r)}) - \eta \left\langle \nabla \ell(\boldsymbol{\Theta}^{(r)}), \nabla \ell(\boldsymbol{\Theta}^{(r)}) + \delta^{(r)} \right\rangle \\
&\quad + \frac{L\eta^2}{2} \|\nabla \ell(\boldsymbol{\Theta}^{(r)}) + \delta^{(r)}\|^2 \\
&= \ell(\boldsymbol{\Theta}^{(r)}) - \eta \|\nabla \ell(\boldsymbol{\Theta}^{(r)})\|^2 - \eta \left\langle \nabla \ell(\boldsymbol{\Theta}^{(r)}), \delta^{(r)} \right\rangle \\
&\quad + \frac{L\eta^2}{2} \|\nabla \ell(\boldsymbol{\Theta}^{(r)})\|^2 + L\eta^2 \langle \nabla \ell(\boldsymbol{\Theta}^{(r)}, \delta^{(r)}) \rangle + \frac{L\eta^2}{2} \|\delta^{(r)}\|^2 \\
&= \ell(\boldsymbol{\Theta}^{(r)}) + (\frac{L\eta^2}{2} - \eta) \|\nabla \ell(\boldsymbol{\Theta}^{(r)})\|^2 + \frac{L\eta^2}{2} \|\delta^{(r)}\|^2 \\
&\quad + (L\eta^2 - \eta) \left\langle \nabla \ell(\boldsymbol{\Theta}^{(r)}), \delta^{(r)} \right\rangle \\
&= \ell(\boldsymbol{\Theta}^{(r)}) + (\frac{L\eta^2}{2} - \eta) \|\nabla \ell(\boldsymbol{\Theta}^{(r)})\|^2 + \frac{L\eta^2}{2} \|\delta^{(r)}\|^2 \\
&\quad + \frac{L\eta^2 - \eta}{2} \left( \|\nabla \ell(\boldsymbol{\Theta}^{(r)})\|^2 + \|\delta^{(r)}\|^2 - \|\nabla \ell(\boldsymbol{\Theta}^{(r)}) + \delta^{(r)}\|^2 \right) \\
&= \ell(\boldsymbol{\Theta}^{(r)}) + (L\eta^2 - \frac{3\eta}{2}) \|\nabla \ell(\boldsymbol{\Theta}^{(r)})\|^2 + (L\eta^2 - \frac{\eta}{2}) \|\delta^{(r)}\|^2 \\
&\quad - \frac{L\eta^2 - \eta}{2} \|\nabla \ell(\boldsymbol{\Theta}^{(r)}) + \delta^{(r)}\|^2
\end{aligned}
\tag{30}
$$

We select $\eta = \frac{1}{L}$, so we have

$$\ell(\boldsymbol{\Theta}^{(r+1)}) \leq \ell(\boldsymbol{\Theta}^{(r)}) - \frac{1}{2L}\|\nabla\ell(\boldsymbol{\Theta}^{(r)})\|^2 + \frac{1}{2L}\|\delta^{(r)}\|^2 \tag{31}$$

Rearranging the above inequality gives

$$\|\nabla\ell(\boldsymbol{\Theta}^{(r)})\|^2 \leq 2L\left(\ell(\boldsymbol{\Theta}^{(r)}) - \ell(\boldsymbol{\Theta}^{(r+1)})\right) + \|\delta^{(r)}\|^2 \tag{32}$$

Then, summing $r$ from 1 to $T$, we have

$$\min_{r \in [T]} \|\nabla\ell(\boldsymbol{\Theta}^{(r)})\|^2 \leq \frac{2L\left(\ell(\boldsymbol{\Theta}^{(1)}) - \ell(\boldsymbol{\Theta}^{(T+1)})\right)}{T} + \frac{1}{T}\sum_{r \in [T]} \|\delta^{(r)}\|^2 \tag{33}$$

**Definition D.17** ($B$-local dissimilarity). The local functions $\ell_c$ are $B$-locally dissimilar at $\boldsymbol{\Theta}$ if

$$\mathbb{E}_c[\|\nabla\ell_c(\boldsymbol{\Theta})\|^2] \leq \|\nabla\ell(\boldsymbol{\Theta})\|^2 B^2.$$

We further define $B(\boldsymbol{\Theta}) = \sqrt{\frac{\mathbb{E}_c[\|\nabla\ell_c(\boldsymbol{\Theta})\|^2]}{\|\nabla\ell(\boldsymbol{\Theta})\|^2}}$ for $\|\nabla\ell(\boldsymbol{\Theta})\| \neq 0$.

**Definition D.18** ($\gamma$-inexact solution). For a function $h(w; w_0) = F(w) + \lambda\|w - w_0\|^2$, and $\gamma \in [0, 1]$, we say $w^*$ is a $\gamma$-inexact solution of $\min_w h(w; w_0)$ if $\|\nabla h(w^*; w_0)\| \leq \gamma\|\nabla h(w_0; w_0)\|$, where $\nabla h(w; w_0) = \nabla F(w) + \mu(w - w_0)$, where, $\mu = 2\lambda$. Note that smaller $\gamma$ corresponds to higher accuracy.

Using the notion of $\gamma$-inexactness for each local client, we can define $e_c^{(r)}$ such that

$$\nabla\ell_c\left(\boldsymbol{\Theta}_c^{(r+1)}\right) + \mu\left(\hat{\boldsymbol{\theta}}_s^{(r)} - \boldsymbol{\theta}_{s_c}^{(r)}\right) + \mu\left(\boldsymbol{\theta}_c^{(r+1)} - \boldsymbol{\theta}_c^{(r)}\right) - e_c^{(r)} = 0,$$
$$\|e_c^{(r)}\| \leq \gamma\|\nabla\ell_c\left(\boldsymbol{\Theta}_c^{(r)}\right)\|. \tag{34}$$

Then we have

$$\boldsymbol{\theta}_s^{(r+1)} - \boldsymbol{\theta}_s^{(r)} = \frac{-1}{\mu}\mathbb{E}_c\left[\nabla\ell_c\left(\boldsymbol{\Theta}_c^{(r)}\right)\right] + \frac{1}{\mu}\mathbb{E}_c[e_c^{(r)}] - \mathbb{E}_c\left[\Delta\boldsymbol{\theta}_c^{(r)}\right], \tag{35}$$

According to (Li et al., 2020a) and triangle inequality, when a regularization is incorporated, ($\lambda > 0$), we have

$$\frac{1}{4\lambda^2}\|\delta^{(r)}\|^2 \leq \left(\mathbb{E}_c\left[\|\boldsymbol{\theta}_s^{(r+1)} - \boldsymbol{\theta}_{s_c}^{(r)}\|\right]\right)^2$$

$$\leq \left(\frac{1+\gamma}{\bar{\mu}}\right)^2\left(\mathbb{E}_c\left[\|\nabla\ell_c\left(\boldsymbol{\Theta}_c^{(r)}\right) - \Delta\boldsymbol{\theta}_c^{(r)}\|\right]\right)^2$$

$$\leq \left(\frac{1+\gamma}{\bar{\mu}}\right)^2\left(\mathbb{E}_c\left[\|\nabla\ell_c\left(\boldsymbol{\Theta}_c^{(r)}\right) - \Delta\boldsymbol{\theta}_c^{(r)}\|^2\right]\right)$$

$$\leq \frac{B^2(1+\gamma)^2}{\bar{\mu}^2}\mathbb{E}\left[\|\nabla\ell_c\left(\boldsymbol{\Theta}_c^{(r)}\right)\|^2\right] + C,$$

Based on the assumption of the bounded gradients (Assumption D.15), we find that the $\delta^{(r)}$ is also bounded. Specifically, $C = \left(\frac{1+\gamma}{\bar{\mu}}\right)^2\mathbb{E}_c[\|\Delta\boldsymbol{\theta}_c\|^2] \approx \left(\frac{1+\gamma}{\bar{\mu}}\right)^2\mathbb{E}[\|\Delta M_c\|^2]$. $\|\delta^{(r)}\|^2$ measures the degree of data heterogeneity.

Overall, when $\lambda = 0$, the term $\delta^{(r)} = 0$, eliminating the impact of data heterogeneity and resulting in a convergence rate of $O\left(\frac{1}{T}\right)$, consistent with FedAvg. And when incorporating regularization ($\lambda > 0$), we establish that $\|\delta^{(r)}\|^2$ is bounded, analogous to the theoretical guarantees provided by FedProx (Li et al., 2020a).

This analysis suggests that FlatLand can balance the trade-off between computational overhead and model effectiveness as the number of clients increases. While it introduces additional operations in local computation, these overheads are limited and offer optimization opportunities through pre-computation and caching. The method compensates for these costs through reduced communication overhead and enhanced representation capability in Lorentz space, making it a practical choice for personalized federated learning.

## D.5. Perspectives on Lorentz Transformations

Lorentz Boosts and Lorentz Rotations (Appendix B.3) are understood as transformations that are encapsulated by $\text{LT}\left(\mathbf{x}; \hat{\mathbf{M}}\right)$ when the dimension is unchanged (Chen et al., 2021). We can easily prove that the Lorentz transformations are still covered by $\text{LT}\left(\cdot; \Phi\left(\hat{\mathbf{M}}, \mathbf{N}\right)\right)$, where the corresponding Lorentz-linear block is $\hat{\mathbf{M}} = [\mathbf{m}\ \mathbf{M}] \in \mathbb{R}^{n \times (n+1)}$ and the replacement block satisfies $\mathbf{N} \in \mathbb{R}^{n \times n}$. For any data point $\mathbf{x} \in \mathcal{D}_c$, transformations $\text{LT}\left(\mathbf{x}; \hat{\mathbf{M}}\right)$ and $\text{LT}\left(\mathbf{x}; \Phi\left(\hat{\mathbf{M}}, \mathbf{N}\right)\right)$ map $\mathbf{x}$ to new Lorentz-model coordinates while preserving the Lorentzian norm constraint (Proposition 6.1). Thus, replacing the shared block after aggregation keeps the representation on the same client-specific hyperboloid. Clients with different Lorentz scale parameters $K_c$ therefore remain in distinct Lorentz spaces, reflecting differences in their underlying data distributions. Moreover, according to the definition of Lorentz rotation in Equation (8), the server updates only $\mathbf{M}$ while leaving the time-like component local. This operation is a relaxation of a Lorentz rotation, consistent with our "Flatland" assumption that aggregates only space-like information.

# E. Experimental Supplementary

## E.1. Datasets

For federated node classification, we adopt four benchmark datasets constructed by (Baek et al., 2023): Cora, CiteSeer, ogbn-arxiv, and Photo (Sen et al., 2008; Hu et al., 2020; Shchur et al., 2018). Cora, CiteSeer, and ogbn-arxiv are citation graphs. Photo is a product graph. Each graph dataset is divided into a certain number of disjoint subgraphs using the METIS graph partitioning algorithm (Karypis & Kumar, 1995), where each subgraph belongs to an FL client. Statistics of datasets are summarized in Table 6.

For federated graph classification, we consider the non-IID settings proposed by (Xie et al., 2021). In total, there are 13 graph classification datasets from three different domains, including small molecules (MUTAG, BZR, COX2, DHFR, PTC_MR, AIDS, NCI1) denoted as CHEM, bioinformatics (ENZYMES, DD, PROTEINS) denoted as BIO, and social networks (COLLAB, IMDB-BINARY, IMDB-MULTI) (Morris et al., 2020) denoted as SN. To simulate data heterogeneity, two non-IID settings are constructed: (1) a cross-dataset setting based on the small molecule datasets (CHEM), (2) a cross-domain setting based on all datasets (BIO-CHEM-SN). In each setting, one dataset corresponds to one FL client. Statistics of datasets are summarized in Table 5 and Table 7.

*Table 5.* Statistics of graph classification datasets. We report the (average) number of graphs, nodes, edges, classes, and node features of each dataset.

| Dataset | CHEM | | | | | | | BIO | | | SN | | |
|---|---|---|---|---|---|---|---|---|---|---|---|---|---|
| | MUTAG | BZR | COX2 | DHFR | PTC_MR | AIDS | NCI1 | ENZYMES | DD | PROTEINS | COLLAB | IMDB-BINARY | IMDB-MULTI |
| # Graphs | 188 | 405 | 467 | 467 | 344 | 2000 | 4110 | 600 | 1178 | 1113 | 5000 | 1000 | 1500 |
| Avg. # Nodes | 17.93 | 35.75 | 41.22 | 42.43 | 14.29 | 15.69 | 29.87 | 32.63 | 284.32 | 39.06 | 74.49 | 19.77 | 13.00 |
| Avg. # Edges | 19.79 | 38.36 | 43.45 | 44.54 | 14.69 | 16.20 | 32.30 | 62.14 | 715.66 | 72.82 | 2457.78 | 96.53 | 65.94 |
| # Classes | 2 | 2 | 2 | 2 | 2 | 2 | 2 | 6 | 2 | 2 | 3 | 2 | 3 |
| Node Features | original | original | original | original | original | original | original | original | original | original | degree | degree | degree |

*Table 6.* Statistics of homophilic node classification datasets. We report the (average) number of nodes, edges, classes, clustering coefficient, and heterogeneity for different numbers of clients.

| Dataset | Cora | | | CiteSeer | | | ogbn-arxiv | | | Amazon-Photo | | |
|---|---|---|---|---|---|---|---|---|---|---|---|---|
| # Clients | 1 | 10 | 20 | 1 | 10 | 20 | 1 | 10 | 20 | 1 | 10 | 20 |
| # Classes | | 7 | | | 6 | | | 40 | | | 8 | |
| Avg. # Nodes | 2,485 | 249 | 124 | 2,120 | 212 | 106 | 169,343 | 16,934 | 8,467 | 7,487 | 749 | 374 |
| Avg. # Edges | 10,138 | 891 | 422 | 7,358 | 675 | 326 | 2,315,598 | 182,226 | 86,755 | 238,086 | 19,322 | 8,547 |
| Avg. Clustering Coefficient | 0.238 | 0.259 | 0.263 | 0.170 | 0.178 | 0.180 | 0.226 | 0.259 | 0.269 | 0.410 | 0.457 | 0.477 |
| Heterogeneity | N/A | 0.606 | 0.665 | N/A | 0.541 | 0.568 | N/A | 0.615 | 0.637 | N/A | 0.681 | 0.751 |

*Table 7.* Statistics of heterophilic node classification datasets. We report the (average) number of nodes, edges, and classes.

| Dataset | Roman-empire | | | Minesweeper | | | Tolokers | | | Questions | | |
|---|---|---|---|---|---|---|---|---|---|---|---|---|
| # Clients | 1 | 10 | 20 | 1 | 10 | 20 | 1 | 10 | 20 | 1 | 10 | 20 |
| # Classes | | 18 | | | 2 | | | 2 | | | 2 | |
| Avg. # Nodes | 22,662 | 2,326 | 1,124 | 10,000 | 1,008 | 513 | 11,758 | 1,138 | 605 | 48,921 | 5,037 | 2,484 |
| Avg. # Edges | 32,927 | 6,682 | 3,268 | 39,402 | 7,696 | 3,820 | 519,000 | 40,000 | 15,000 | 153,540 | 20,000 | 7,702 |

## E.2. Implementation Details

**Implementation of learnable curvature.** For each client $c$, we optimize a raw scalar $\kappa_c$ and map it to a positive Lorentz scale parameter $K_c = \text{sigmoid}(\kappa_c) + 0.5$. This keeps $K_c$ in the effective range $[0.5, 1.5]$, so the corresponding sectional curvature $-1/K_c$ remains negative while numerical optimization stays stable (Chen et al., 2021). The reparameterization allows the corresponding curvature to adapt to heterogeneous client data during local training.

**Implementation of node classification and graph classification tasks.** For node classification, we use a 2-layer GCN (Kipf & Welling, 2017) for Euclidean models, a 2-layer LGCN (Chen et al., 2021) for FlatLand, and HGCN with node selection for FedHGCN (Du et al., 2024). LGCN serves as the backbone for our graph learning framework, combining

Lorentz linear layers with graph aggregation operations, similar to how Euclidean counterparts such as GCN and GIN integrate linear layers with graph aggregation. Each layer applies a Lorentz transformation followed by neighbor aggregation using the adjacency matrix to obtain node representations. We conduct 100 rounds for Cora/CiteSeer and 200 rounds for larger datasets such as Photo and ogbn-arxiv, with 1–3 local epochs and 128-dimensional hidden layers. For graph classification, we use a 3-layer GIN (Xu et al., 2018) as the Euclidean encoder and the same 3-layer hyperbolic encoders as in node classification for hyperbolic models, with 1 local epoch and 200 rounds. The learning rate is chosen from $\{0.01, 0.001\}$, and the weight decay is $10^{-5}$. We optimize with Adam and report node- or graph-level accuracy averaged across clients. All experiments are implemented in Python 3.10 and PyTorch and run on an RTX A6000 GPU. Each client is allocated a worker; one node-classification local epoch takes roughly one second per round.

**Backbone structure.** Inspired by recent GNN models (Chen et al., 2024) that highlight the benefits of incorporating high-pass information for heterophilic graphs, all hyperbolic baselines in our framework combine information from both low-pass (adjacency-based) and high-pass (Laplacian-based) operations through a learnable gating mechanism. For a fair comparison, we also applied this design to the Euclidean baselines, but it did not yield improvements over the hyperbolic counterparts, suggesting that hyperbolic models benefit more than Euclidean baselines from jointly using low-pass and high-pass information for heterophilic graph representation.

### E.3. Baseline Selection and Comparison

We provide clarification on our baseline selection to ensure fair and comprehensive comparison:

**Euclidean baselines.** We compare against representative methods from each major PFL category: (1) *basic FL*: FedAvg (McMahan et al., 2017); (2) *regularization-based*: FedProx (Li et al., 2020a); (3) *parameter decoupling*: FedPer (Arivazhagan et al., 2019); and (4) *client clustering*: GCFL (Xie et al., 2021). These baselines span the spectrum of PFL approaches and use the same GNN backbone (GCN/GIN) as our method for fair comparison.

**Hyperbolic baseline.** FedHGCN (Du et al., 2024) is the most relevant hyperbolic FL baseline, combining FedAvg with hyperbolic GNNs. Unlike FlatLand, FedHGCN uses a fixed global curvature and lacks personalization capability, making it a direct comparison point for demonstrating the benefits of tailored curvature and parameter decoupling.

**Local training baselines.** We include Local ($E$) and Local ($L$) to show the performance of purely local training in Euclidean and Lorentz spaces respectively, without any federated aggregation. This helps isolate the contribution of federated learning from the geometric representation.

**Why not other hyperbolic models?** Methods like HyperFed (Liao et al., 2023) and FedMRUR (An et al., 2023) focus on different aspects (prototype learning and knowledge distillation) rather than personalized geometric modeling. Our comparison therefore focuses on methods that directly address the heterogeneity challenge through model architecture or aggregation designs that explicitly handle client heterogeneity.

### E.4. Unified Summary of Ablations

The main text and appendices evaluate the components of FlatLand from different angles. We collect these results in Table 8 to clarify which design choice each ablation isolates.

Overall, the gains do not come from a single factor. Local ($L$) versus Local ($E$) shows that Lorentz representations are beneficial for many graph clients, while the gap between FlatLand and w/o DS highlights the importance of separating personalized time-like parameters from shared space-like parameters. The drop of w/o TS further indicates that client-specific curvature is useful, whereas the similar performance across curvature initializers suggests that learnable curvature matters more than a particular initialization rule.

### E.5. Impact of Curvature Initialization

We further investigate the impact of different curvature initialization strategies on model performance. Specifically, we compare four initialization methods: (1) **Forman-Ricci curvature**, (2) **Ollivier-Ricci curvature**, (3) **constant-$K = 1$**, and (4) an **MLP-based** estimator that updates the Lorentz scale parameter with an MLP layer. Table 9 reports the corresponding

*Table 8.* Unified ablation summary on Cora and CiteSeer with 20 clients. Results are reported as mean accuracy.

| Group | Variant | Cora | CiteSeer | Source |
|---|---|---|---|---|
| Full model | FlatLand | 82.49 | 72.24 | Sec. 7.2, Table 1 |
| Representation | Local ($E$) | 80.30 | 65.98 | Sec. 7.2, Table 1 |
| Representation | Local ($L$) | 80.46 | 69.52 | Sec. 7.2, Table 1 |
| Decoupling | FlatLand ($E$) | 76.23 | 66.29 | Sec. 7.5, Table 4 |
| Decoupling | w/o DS | 71.76 | 63.08 | Sec. 7.5, Figure 5 |
| Tailored curvature | w/o TS | 78.83 | 67.93 | Sec. 7.5, Figure 5 |
| Curvature initialization | Ricci | 82.49 | 72.24 | App. E.5, Table 9 |
| Curvature initialization | Constant | 81.91 | 71.89 | App. E.5, Table 9 |
| Curvature initialization | Ollivier | 82.51 | 72.21 | App. E.5, Table 9 |
| Curvature initialization | MLP | 82.33 | 72.59 | App. E.5, Table 9 |

results for CiteSeer under the 10-client and 20-client settings.

*Table 9.* Performance with different curvature initialization methods on CiteSeer. Results are reported as mean $\pm$ standard deviation over five runs.

| Clients | Ricci | Ollivier | Const. | MLP |
|---|---|---|---|---|
| 10 | $73.90 \pm 0.23$ | $73.51 \pm 0.18$ | $72.91 \pm 0.20$ | $72.96 \pm 0.46$ |
| 20 | $72.24 \pm 0.24$ | $72.21 \pm 0.26$ | $71.89 \pm 0.31$ | $72.59 \pm 0.50$ |

As shown in Table 9, the choice of initialization method has only a marginal effect on performance. This indicates that our method is robust to the initialization choice, and that the specific initializer is *not critical* to the effectiveness of our method. What truly matters is that each client is assigned a **learnable curvature**, which can be adapted during training to better fit its local data distribution (see Figure 5).

### E.6. Convergence Curves

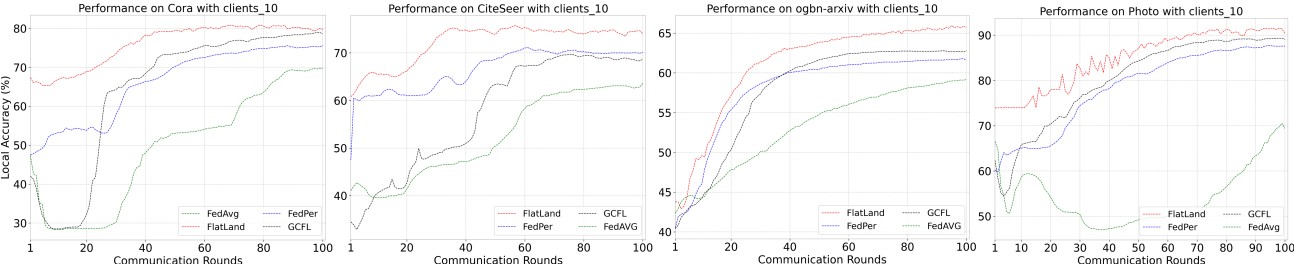

*Figure 10.* Convergence curves of FlatLand and strong baselines.

The convergence curves are shown in Figure 10. As the figures demonstrate, our proposed method achieves competitive convergence speed while reaching higher final performance. This is consistent with the theoretical discussion in Appendix D.4: the decoupling strategy does not introduce additional client-similarity estimation or clustering optimization, and the server-side aggregation remains as simple as averaging the shared space-like parameters. Therefore, FlatLand can retain FedAvg-level convergence behavior while better handling heterogeneous local geometries.

### E.7. Curvature Sensitivity and Interpretation

To further clarify why curvature should be client-specific and learnable, we summarize the relevant empirical observations:

**Client-wise preferred curvature.** Different sources of client heterogeneity, including graph-structural variation (e.g., degree distribution, clustering patterns, and homophily) and label-distribution shift, may induce different preferred geometric biases across clients. Empirically, we observe that different clients can achieve their best validation performance under different curvature values, rather than sharing a single globally optimal curvature or consistently preferring the Euclidean limit. This suggests that client-specific curvature acts as a compact geometric proxy for heterogeneous client conditions,

instead of merely serving as a numerical scaling factor. This also explains why directly fixing client curvature from a structure-only Ricci estimate is insufficient. The estimated Ricci curvature provides an informative initialization signal, but the curvature that benefits downstream prediction is also shaped by label-distribution shift, feature geometry, and task-specific optimization. Therefore, FlatLand treats curvature as a learnable client-specific parameter: the structural estimate serves as a prior, while training can adapt it toward the effective geometry preferred by each client.

**Fixed vs. learnable curvature.** As shown in Table 9 and the ablation study in the main text, fixing curvature to a constant value consistently underperforms learnable curvature, with performance gaps of 1–3% across datasets. This demonstrates the importance of client-specific geometric adaptation.

