# OpenReview forum: "FlatLand: Personalized Graph Federated Learning via Tailored Lorentz Space"
_ICML.cc/2026/Conference — ICML 2026 spotlight_

### Official Review · Reviewer_L7WS · 2026-03-10

**Soundness:** 3
**Presentation:** 4
**Significance:** 3
**Originality:** 3
**Overall Recommendation:** 5
**Confidence:** 3

**Summary:**

This paper focuses on handling graph heterogeneity in Personalized Federated Learning (PFL). The authors propose FlatLand, which maps data from different clients into tailored Lorentz spaces. Overall, the paper is logically structured, the standpoints are well supported, and the experiments are extensive. However, several issues remain: some concepts are not clearly explained, the applicable scope of the method is not well clarified, and certain formulas lack sufficient interpretation. Nevertheless, these issues do not overshadow the main contributions of the work, and therefore I lean toward acceptance.

**Compliance With Llm Reviewing Policy:**

Affirmed.

**Key Questions For Authors:**

1. Graph curvature vs. hyperbolic curvature. The paper uses Forman-Ricci curvature of the graph as a proxy for the curvature in hyperbolic space. Since these two notions of curvature are conceptually different, could the authors clarify the theoretical or empirical justification behind this choice?
2. Choice of Forman-Ricci curvature. Why does the method rely on Forman-Ricci curvature instead of Ollivier-Ricci curvature, which is often considered more closely related to graph geometry? What advantages does Forman-Ricci provide in the proposed framework?
3. Applicability to different graph structures. Hyperbolic representations are typically well suited for tree-like or hierarchical graphs. Could the authors further discuss what types of graph structures are expected to benefit most from the proposed method, and why the method appears to perform well across several datasets with potentially different structural properties?

**Limitations:**

1. Although FlatLand demonstrates strong performance across multiple datasets, several limitations remain. First, it is still unclear which types of graph structures are most suitable for this method. In particular, hyperbolic representations are typically advantageous for hierarchical or tree-like graphs, but the paper does not provide sufficient analysis of how different structural properties of graphs affect the performance of the proposed approach.
2. The method may implicitly rely on certain data distribution characteristics across clients, but the paper does not analyze whether specific client-level distribution patterns are required for the method to work effectively.
3. The method shows stronger improvements on node classification tasks than on graph classification tasks. The underlying reason for this discrepancy is not discussed in the paper, and further analysis would help clarify when and why the proposed approach is most beneficial.

**Strengths And Weaknesses:**

Strength
1. The paper proposes FlatLand, a simple yet effective method for graph learning under PFL, which essentially extends FedAvg into hyperbolic space.
2. The paper provides substantial theoretical analysis, along with empirical validation, to demonstrate the relationship between curvature and the Lorentz space.
3. Extensive and comprehensive experiments are conducted to verify the effectiveness of the proposed method.
4. The paper is generally well written, with clear logical flow and presentation.

Weaknesses
1. Unclear explanation of the parameter decoupling strategy. The paper claims that the proposed strategy separates time-like and space-like parameters and treats the time-like parameters as personalized ones. However, according to Equation (4), the time-like parameters at layer l+1are derived from both the time-like and space-like parameters at layer l. This seems inconsistent with the claim of separation and would benefit from clearer explanation.
2. Ablation study design could be improved. The current ablation study compares local(E) and local(L), which does not clearly isolate the contributions of the key design choices. A more informative analysis would directly evaluate the effects of the two main components: Curvature Estimation and the Parameter Decoupling Strategy. For example, Curvature Estimation could be evaluated by fixing the initial curvature (rather than forcing it to be all −1), while Parameter Decoupling Strategy could be evaluated by removing the proposed decoupling mechanism.
3. Code availability. The repository link appears to be outdated or inaccessible, which makes it difficult to reproduce the results.

---

> ### Author Rebuttal · Authors · 2026-03-31
>
> We sincerely thank the reviewer for the positive comments. Below, we use **W/Q** for weaknesses/questions and **R** for our responses
>
> **W1: Unclear explanation of the parameter decoupling strategy.**
>
> **R:** Thank you for the comment. We will clarify this in our revision paper. Our decoupling strategy focuses on separating the parameters at each layer of the model. As discussed in Theorem 3.3, the key component that captures data heterogeneity is **the time-like dimension $x_t$ of the hyperbolic embedding**. After each layer of the Lorentz neural network, the input vectors are transformed into a new hyperbolic embedding, with the time-like dimension updated as
> $
> x_t^{(l+1)} = \sqrt{\left\| m x_t^{(l)} + \mathbf{M} x_s^{(l)} \right\|^2 + K}.
> $
> This means that the component at layer $l+1$ that carries the heterogeneity information is transferred as $x_t^{(l+1)}$. Thus, the parameters associated with $x_t^{(l+1)}$ are naturally aligned with the heterogeneity information. This ensures that our decoupling strategy is consistent with the theoretical derivation and can be effectively applied across multiple layers.
>
> **W2: Ablation study.**
>
> **R:** Thanks very much for the suggestion. We agree that the relevant evidence is currently scattered across different sections and appendix results, and should be discussed in a more unified manner. To make this clearer, we summarize the key findings below together with the corresponding sections, tables, and figures where the detailed analyses are provided in [**Anonymous Table**](https://anonymous.4open.science/w/flatlands-F989/?space=space-c&item=table-aczz-w2-2-overall-performance).
>
> **Overall takeaway.** Our ablations suggest that the performance gain comes from the combination of three factors rather than a single source.
>
> (1) Role of hyperbolic geometry. Local (L) and Local (E) show that hyperbolic representation itself already brings gains in most of the datasets; while they are not always the best-performing variants, they remain competitive, and the remaining gap is mainly due to the additional benefit of the decoupling strategy.
>
> (2) Role of parameter strategy. The degradation of *w/o DS* confirms that parameter decoupling is another key factor, providing additional gains beyond hyperbolic representation alone.
>
> (3) Role of client-specific curvature. The performance drop of *w/o TS* confirms the importance of client-specific curvature, while the similar results across different curvature initializations suggest that the key is the ability to learn suitable curvature for each client, rather than any specific initializer.
>
> **W3: Code availability.**
>
> **R:** Thanks for pointing this out. We have renewed the anonymous code link.
>
> **Q1 & Q2: Graph curvature vs. hyperbolic curvature.**
>
> **R:** Thanks for the question. Our theoretical connection is well supported by two complementary lines of prior work. First, [1-3] theoretically and empirically show that hyperbolic geometry is not an ad hoc modeling choice for complex graphs: hyperbolic negative curvature naturally gives rise to heterogeneous degree distributions and discrete curvature.  Second, **Forman-Ricci and Ollivier-Ricci are both discrete analogues of classical Ricci curvature**, and prior empirical studies show that they are often strongly correlated across many synthetic and real networks, including hyperbolic geometric graphs. Mainly, the choice of the curvature serves as an analysis tool; in practice, **our method is fairly robust to the initialization choice**, so any reasonable initializer works.
>
> **Q3: Applicability to different graph structures.**
>
> **R:** Thank you for the question. We agree that the benefit of hyperbolic modeling depends on whether the data exhibits suitable geometry. Empirically, across the datasets, most of the `Local (L)` often outperforms its Euclidean counterpart `Local (E)`, indicating that hyperbolic representations are beneficial in many graph settings.
>
> Meanwhile, on some datasets, hyperbolic modeling is not always superior. Nevertheless, FlatLand remains competitive in these cases, since its advantage comes not only from hyperbolic adaptation **but also from the decoupling strategy**, which improves the preservation of local information and the transfer of useful shared knowledge across clients. As a result, even when not all clients benefit equally from hyperbolic geometry, the overall method still performs strongly. We will make this scope clearer in the revision.
>
> We also note that this framework has broader potential: a unified mixed-curvature design may better fit more complex data and further improve performance, which we will discuss in the limitations section.
>
> ---
>
> [1] Hyperbolic Geometry of Complex Networks. 2010.
>
> [2] Hyperbolicity Measures "Democracy" in Real-World Networks. 2025.
>
> [3] Comparative analysis of two discretizations of Ricci curvature for complex networks. 2018.

---

> > ### Author Rebuttal · Reviewer_L7WS · 2026-04-04
> >
> > Thanks for the response and I will keep my positive score.

---

> > > ### Author Response · Authors · 2026-04-04
> > >
> > > We sincerely appreciate your positive feedback and are pleased that our rebuttal has addressed your concerns. We will incorporate the rebuttal into the final version of our manuscript.

---

### Official Review · Reviewer_ACzZ · 2026-03-11

**Soundness:** 3
**Presentation:** 3
**Significance:** 3
**Originality:** 3
**Overall Recommendation:** 4
**Confidence:** 2

**Summary:**

The paper proposes FlatLand, a personalized federated learning (PFL) framework for graph data that operates in client-specific Lorentz (hyperboloid) spaces with learnable curvature. The core idea is to (i) assign each client its own hyperbolic curvature to better match intrinsic geometric properties estimated via discrete Ricci curvature, and (ii) decouple model parameters along hyperbolic time-like (personalized) versus space-like (shared/aggregated) directions. The authors provide theoretical motivation for curvature personalization and time-like heterogeneity encoding, prove that the aggregation preserves the Lorentz manifold constraints, and demonstrate empirical gains on homophilic and heterophilic benchmarks, with especially strong improvements in low-dimensional regimes.

**Compliance With Llm Reviewing Policy:**

Affirmed.

**Final Justification:**

This paper presents a technically solid and well-motivated approach to personalized graph federated learning by introducing a geometric perspective based on Lorentz space and client-specific curvature. I find the method reasonably original, and the overall framework is clearly designed and empirically validated on multiple benchmarks. In particular, the idea of decoupling personalized and shared components through time-like and space-like parameters is interesting and meaningful for modeling structural heterogeneity across clients.

My main concerns remain around the extent to which the proposed geometric mechanism is directly validated, especially the specific role of the time-like dimension and the applicability of the method to graphs without strong hyperbolic structure. That said, the rebuttal addressed my concerns reasonably well.

Overall, I believe the paper makes a worthwhile contribution and I support a weak accept.

**Key Questions For Authors:**

1. Is curvature K learned per client globally or per layer? Did you try layerwise K or a small set of Ks per model, and how did that affect performance and stability?

2. How do you compute Forman-Ricci in large graphs (especially the triangle term)? Do you use approximations or sparse algorithms?

3. Did you experiment with alternative curvature estimators (Ollivier-Ricci approximations, ERC) or with per-edge/cluster curvature that might reduce within-client multimodality? If so, what were the findings?

**Limitations:**

see weakness

**Strengths And Weaknesses:**

**Strengths**

1. The paper is generally easy to follow, and the overall motivation and methodology are clearly presented.

2. Personalization in edge-side federated learning is an important and practical research topic, especially given the growing number of decentralized devices and privacy constraints.

3. Using a graph structure to model relationships among clients is a reasonable design choice.

**Weakness**

1. For theoretical concerns, Theorem 3.1’s distortion lower bound relies on average discrete Ricci curvature and bi-Lipschitz embeddings into L^d_K; the link between average Forman-Ricci on graphs and the optimal curvature of a continuous hyperbolic manifold embedding is not fully justified. Key assumptions and constants are not transparent in the main text. Theorem 3.3’s mutual-information claims rest on stylized assumptions. In practice, the space-like magnitude depends on K via sinh(ρ/√K), so M-aggregation may still face client-scale mismatch.

2. For experiments: Lacks comparisons or ablations on alternative curvature estimators (e.g., Ollivier-Ricci, Effective Resistance Curvature) and sensitivity of learned performance to curvature-estimation noise/bias. On several homophilic datasets, FlatLand is not consistently superior (e.g., Cora-10 and Photo-10 where FED-PUB slightly outperforms), and graph classification gains are modest. This slightly weakens the “across-the-board” superiority claim.

---

> ### Author Rebuttal · Authors · 2026-03-31
>
> We sincerely thank the reviewer for the constructive comments. Below, we use **W/Q** for weaknesses/questions and **R** for our responses.
>
> **W1.1: Forman-Ricci on graphs and the hyperbolic curvature.**
>
> **R:** Thanks. Our theoretical motivation is well supported by two complementary lines of prior work. First, [1, 2] support that hyperbolic geometry is not an ad hoc modeling choice for complex graphs: hyperbolic negative curvature naturally gives rise to heterogeneous degree distributions and discrete curvature.  Second, Forman-Ricci and Ollivier-Ricci are both discrete analogues of classical Ricci curvature, and prior empirical studies show that they are often strongly correlated across many synthetic and real networks, including hyperbolic geometric graphs.
>
> **W1.2: The K-related magnitude of the client.**
>
> **R:** Thanks for the interesting observation. Theorem 3.3 is intended to characterize the dominant carrier of client-specific information, rather than to claim that the space-like part is strictly curvature-invariant in practice. While scaling may appear in the space-like magnitude, *this does not affect the main role of the result: the time-like component is the primary carrier of client-specific heterogeneity.* Accordingly, our method does not require such a mismatch to vanish completely; it only requires the decoupling to isolate the main client-specific factor, which is consistent with the empirical gains over both no decoupling and Euclidean splitting. We agree that a more refined treatment of residual curvature-dependent scaling is an interesting direction for future work, and we will clarify this scope in the revision.
>
> **W2.1 & Q3: Lacks ablations on alternative curvature estimators.**
>
> **R:** Thanks. In App. E.4 (Tab. 8), we study different curvature initializations. The similar results across initializations show that **our method is robust to initialization**; the key is to automatically learn suitable client-specific curvature during training, rather than relying on any specific initializer. This is consistent with the broader practice of learning geometric parameters from data to fit an appropriate hyperbolic structure [3]. We summarize a comprehensive ablation study in [**Anonymous Table**](https://anonymous.4open.science/w/flatlands-F989/?space=space-c&item=table-rmzv-w1-q1-ablation-study) for a more detailed check.
>
> **W2.2:  FlatLand is not consistently superior in all datasets.**
>
> **R:** Thanks. Although FlatLand does not achieve the best result on every individual task, it delivers the strongest **overall** performance: across four aggregated regimes—homo_10, homo_20, hetero_10, and hetero_20 (16 tasks in total)—FlatLand ranks first with over 6% improvement for the second best in [**Anonymous Table**](https://anonymous.4open.science/w/flatlands-F989/?space=space-c&item=table-aczz-w2-2-overall-performance).
>
> Meanwhile, FlatLand remains simple and efficient. Unlike strong baselines that rely on explicit parameter-similarity computation or auxiliary models during aggregation, FlatLand uses only **parameter decoupling**, yet still substantially outperforms other decoupling-based methods (e.g., **+21.1** over FedPer on the overall average).
>
> More importantly, to the best of our knowledge, this is the **first** work to study personalized federated graph learning from a geometric perspective. We therefore adopt a simple framework to validate this idea. We believe the current results already demonstrate the promise of this direction, while also leaving room for future improvement; for example, a unified mixed-curvature design combining hyperbolic, Euclidean, and spherical geometry may better fit more complex data.
>
> **Q1:  Layerwise K or a small set of Ks.**
>
> **R:**  Curvature is a global learnable scalar per client, rather than per layer. As a lightweight check, we also tested a layerwise variant with multiple curvature parameters, which yields modest gains on both Cora and CiteSeer. This suggests that finer-grained curvature parameterization may be beneficial. We thank the reviewer for this insightful suggestion; however, it is more related to Lorentz neural network architecture design than to the main focus of this paper, and we leave a systematic study to future work.
>
> | Dataset | Global K | Layerwise K |
> | --- | --- | --- |
> | Cora | 66.24 | 68.24 |
> | CiteSeer | 64.03 | 64.75 |
>
> **Q2:  How to compute the Forman-Ricci in large graphs?**
>
> **R:** Thanks for the question. We compute Forman-Ricci curvature using the `GraphRicciCurvature` package. This is estimated once from the input graph before training. Therefore, the practical overhead is usually modest. We do not use additional approximations beyond the package implementation.
>
> ---
>
> [1] Hyperbolic Geometry of Complex Networks. 2010.
>
> [2] Comparative analysis of two discretizations of Ricci curvature for complex networks. 2018.
>
> [3] A self-supervised Riemannian GNN with time-varying curvature for temporal graph learning. 2022.

---

> > ### Author Rebuttal · Reviewer_ACzZ · 2026-04-03
> >
> > Thanks to the authors for their effort in the rebuttal. My concerns have been well addressed, and I will raise my score.

---

> > > ### Author Response · Authors · 2026-04-03
> > >
> > > Thank you very much for your kind acknowledgement. We are glad that our rebuttal has addressed your concerns, and we will refine the paper accordingly.

---

### Official Review · Reviewer_SdJM · 2026-03-12

**Soundness:** 3
**Presentation:** 3
**Significance:** 3
**Originality:** 3
**Overall Recommendation:** 5
**Confidence:** 3

**Summary:**

This paper proposes FlatLand, a personalized graph federated learning method that leverages tailored Lorentz space to model structural heterogeneity across clients. The key idea is that many real-world graphs exhibit negative curvature, while different clients may possess graph data with different intrinsic geometric properties. Based on this observation, the method assigns each client its own hyperbolic curvature and performs learning in client-specific Lorentz spaces. A central design is the decoupling of model parameters into personalized time-like parameters and shared space-like parameters, so that only the latter are aggregated on the server. The paper presents theoretical motivation for this geometric formulation and demonstrates through experiments on several node and graph classification benchmarks that the proposed method achieves competitive or superior performance compared with existing personalized federated learning and federated graph learning baselines, with particularly strong results in heterogeneous and low-dimensional scenarios.

**Compliance With Llm Reviewing Policy:**

Affirmed.

**Key Questions For Authors:**

1. The design of FlatLand relies on the assumption that graph data exhibits clear hyperbolic or negative curvature characteristics. For datasets that do not exhibit strong hyperbolic structures, does the proposed method still maintain its advantages? The authors may consider further discussing the applicability of the method across different types of graph structures.

2. One of the core motivations of the method is that client graphs may have different Ricci curvatures. Could the authors further analyze the relationship between the learned curvature and graph structural statistics (e.g., degree distribution, homophily), in order to better clarify the role that curvature plays in the model?

**Limitations:**

The main limitation of this work is that, although the proposed geometric perspective is interesting and the empirical results are generally competitive, the paper does not yet fully validate the claimed mechanism behind the method. In particular, the connection between client heterogeneity and the time-like dimension in Lorentz space is mainly supported by theoretical motivation and indirect ablation results, rather than more direct empirical evidence. In addition, the method appears to be most naturally motivated for graph data with clear hyperbolic or negative-curvature characteristics, while its applicability to datasets without strong non-Euclidean structure is less clear. Finally, although FlatLand outperforms several baselines on many benchmarks, the gains over strong personalized federated learning methods are sometimes moderate, leaving some room for further strengthening the empirical analysis.

**Strengths And Weaknesses:**

Strengths

1. Addresses structural heterogeneity in personalized graph federated learning, which is an important and practical problem.

2. Introduces a geometric perspective by leveraging Lorentz space and hyperbolic geometry for personalized federated graph learning.

3. Proposes a clear parameter decoupling strategy using time-like and space-like dimensions to model personalization and shared knowledge.

4. Evaluates the method on multiple graph learning benchmarks with generally competitive results.

5. The paper is well organized, with coherent connections between motivation, method, and experiments.

Weaknesses

1. The relationship between the time-like dimension and client heterogeneity is not fully validated empirically.

2. The theoretical analysis mainly provides motivation rather than a strict explanation of the performance gains.

3. Performance improvements over strong baselines are sometimes modest on certain datasets.

4. The applicability of the method to datasets without clear hyperbolic structure is not fully discussed.

---

> ### Author Rebuttal · Authors · 2026-03-31
>
> We sincerely thank the reviewer for the constructive comments and positive feedback. Below, we use **W/Q** for weaknesses/questions and **R** for our responses.
>
> **W1 & Q2:  Curvature, time-like dimension, and the heterogeneity.**
>
> **R:**  Thanks. We agree that directly quantifying real-world client heterogeneity is challenging, since common graph statistics (e.g., homophily or degree distribution) only partially characterize structural variation. To directly assess the role of curvature, we directly analyze client-level downstream performance under different curvatures. As shown in [Anonymous Figure](https://anonymous.4open.science/w/flatlands-F989/?space=space-b&item=figure-sdjm-w1-q2-curvature-client-performance), different clients attain their best performance at different curvatures, often distinct from the Euclidean case, suggesting that client heterogeneity manifests as **different preferred geometric biases**. This provides empirical support that curvature captures client-specific structural variation relevant to personalization. The ablations further indicate that the gain comes from the time-/space-like decomposition rather than an arbitrary parameter split.
>
> In our framework, **the time-like dimension preserves the client-specific information induced by this geometric variation.** This is supported by prior work [1,2], which links negative curvature and discrete Ricci curvatures to heterogeneous graph structure. Building on this, Theorem 3.3 and Remark 3.4 formally justify the role of the time-like dimension: under varying client curvatures, it retains mutual information with client identity, whereas the space-like part does not add client-specific information. We will clarify this connection and add the empirical analysis in the revision.
>
> **W2: The theoretical analysis mainly provides motivation rather than a strict explanation of the performance gains.**
>
> **R:** Thanks for this thoughtful comment. We agree that our theory is not intended to directly prove downstream accuracy gains, and we will revise the wording to make this scope clearer. We believe it goes *beyond high-level motivation* by providing a principled justification for our main design choices: Theorem 3.1 motivates client-specific curvature under heterogeneous client geometry; Theorem 3.3 explains why the time-like component carries client-specific heterogeneity; and Theorem/Proposition 5.1 with the convergence analysis ensures that the proposed decoupling and aggregation are geometrically valid and stable. The resulting performance gains are supported empirically by the main results and ablations.
>
> **W3:** **Modest gain on certain datasets.**
>
> **R:** Thanks. While FlatLand is not best on every individual task, it achieves the strongest **overall** performance: across four aggregated regimes—homo_10, homo_20, hetero_10, and hetero_20 (16 tasks total)—it ranks first on all four splits and on the overall average in **[Anonymous Table](**https://anonymous.4open.science/w/flatlands-F989/?space=space-b&item=table-aczz-w2-2-overall-performance**)**.
>
> Meanwhile, FlatLand remains simple and efficient. Unlike other strong baselines that require explicit parameter-similarity computation or auxiliary models during aggregation, FlatLand uses only **parameter decoupling**, yet still clearly outperforms other decoupling-based methods (e.g., **+21.1** over FedPer on the overall average).
>
> To the best of our knowledge, this is the **first** work to study PFGL from a geometric perspective. We therefore intentionally adopt a simple and clean framework to validate this idea. We believe the current results already demonstrate the promise of this direction, while leaving room for further improvement in future work.
>
> **W4 & Q1: Applicability across different graph structures.**
>
> **R:** Thanks for the question. We agree that the benefit of hyperbolic modeling depends on the underlying graph geometry. Empirically, across our datasets, `Local (L)` often outperforms its Euclidean counterpart  `Local (E)`, suggesting that hyperbolic representations are beneficial in many graph settings.
>
> Hyperbolic modeling is not always superior on every dataset. Importantly, FlatLand remains competitive in such cases because its gains come not only from hyperbolic adaptation but also from the decoupling strategy, which better preserves local information while enabling useful cross-client knowledge transfer. Thus, even when some clients benefit less from hyperbolic geometry, the overall method still performs strongly. We will clarify this scope in the revision.
>
> More broadly, this framework **has further potential**: a unified mixed-curvature design may better capture more complex data geometry and further improve performance, which we will discuss in the limitations section.
>
> ---
> [1] Hyperbolic Geometry of Complex Networks. 2010.
>
> [2] Comparative analysis of two discretizations of Ricci curvature for complex networks. 2018.

---

> > ### Author Rebuttal · Reviewer_SdJM · 2026-04-06
> >
> > All of my issues have been resolved, and I have decided to keep my score.

---

> > > ### Author Response · Authors · 2026-04-07
> > >
> > > Thank you very much for your kind acknowledgement. We are glad that our rebuttal has addressed your concerns, and we will incorporate the rebuttal into the final version of our manuscript.

---

### Official Review · Reviewer_rmZV · 2026-03-13

**Soundness:** 3
**Presentation:** 3
**Significance:** 3
**Originality:** 3
**Overall Recommendation:** 4
**Confidence:** 4

**Summary:**

This paper proposes FlatLand, a method for personalized federated graph learning. The motivation is that client graphs may exhibit different structural properties and curvature, which are difficult to capture with traditional Euclidean-based federated learning methods. To address this, FlatLand embeds each client’s data into a tailored Lorentz hyperbolic space with a learnable client-specific curvature parameter. It further exploits the time-like and space-like dimensions in Lorentz space to separate model parameters into personalized and shared components, where only the shared parameters are aggregated during federated training. Experiments on multiple graph datasets and tasks show that the proposed method achieves competitive performance compared with several federated learning baselines, particularly in settings with noticeable client heterogeneity.

**Compliance With Llm Reviewing Policy:**

Affirmed.

**Key Questions For Authors:**

1. Could the authors provide more detailed ablation studies to better understand the contribution of each component of the method?
In particular, it would be useful to more clearly separate the effects of hyperbolic representation, tailored curvature, and parameter decoupling.
2. It would strengthen the paper to include experiments on larger-scale graphs (e.g., ogbn-products) and settings with a larger number of clients.

**Strengths And Weaknesses:**

Strengths:
1. The paper revisits personalized federated graph learning from a geometric perspective by introducing Lorentz hyperbolic space to model structural differences across client graphs. This perspective differs from conventional Euclidean-based approaches and provides a new way to capture structural heterogeneity among clients.
2. The method leverages the structural properties of time-like and space-like dimensions in Lorentz space to naturally separate model parameters into personalized and shared components. This design provides a geometric interpretation for disentangling personalized and shared information, offering a clearer structural motivation compared to some heuristic personalization strategies.
3. The paper provides several theoretical analyses, including motivations relating graph curvature to structural heterogeneity and proofs that the proposed parameter decoupling preserves the validity of the Lorentz manifold after aggregation. These results offer partial theoretical support for the proposed design.

Weakness:
1. The source of the performance improvement is not yet fully disentangled.
It remains unclear whether the observed gains mainly come from the representational advantage of hyperbolic space, the use of client-specific curvature, or the proposed parameter decoupling strategy. Although the paper includes ablation studies, they do not yet fully isolate the contribution of these different factors.
2. The scalability of FlatLand still needs further validation under large-scale graph and large-scale client settings.

---

> ### Author Rebuttal · Authors · 2026-03-31
>
> We sincerely thank the reviewer for the constructive comments and positive feedback. Below, we use **W/Q** for weaknesses/questions and **R** for our responses.
>
> **W1 & Q1: Ablation study for each contribution.**
>
> **R:** Thanks very much for the suggestion. We agree that the relevant evidence is currently scattered across different sections and appendix results, and should be discussed in a more unified manner. To make this clearer, we summarize the key findings below together with the corresponding sections, tables, and figures where the detailed analyses are provided.
>
> | Group | Variant | Cora(20) | CiteSeer(20) | Section / Tab. / Figure |
> | --- | --- | --- | --- | --- |
> |  | `FlatLand(ours)` | **82.49** | **72.24** | Sec. 6.2, Tab. 2 |
> | Representation | `Local(E)` | 80.30 | 65.98 | Sec. 6.2, Tab. 2 |
> | Representation | `Local(L)` | 80.46 | 69.52 | Sec. 6.2, Tab. 2 |
> | Decoupling | `FlatLand(E)` | 76.23 | 66.29 | Sec. 6.4, Tab. 4 |
> | Decoupling | `w/o DS` | 71.76 | 63.08 | Sec. 6.4, Fig. 5 |
> | Tailored curvature | `w/o TS` | 78.83 | 67.93 | Sec. 6.4, Fig. 5 |
> | Curvature robustness | `Ricci` | 82.49 | 72.24 | App. E.4, Tab. 8 |
> | Curvature robustness | `Constant` | 81.91 | 71.89 | App. E.4, Tab. 8 |
> | Curvature robustness | `Ollivier` | 82.51 | 72.21 | App. E.4, Tab. 8 |
> | Curvature robustness | `MLP` | 82.33 | 72.59 | App. E.4, Tab. 8 |
>
> **Overall takeaway.** Our ablations suggest that the performance gain comes from the combination of three factors rather than a single source.
>
> * **Role of hyperbolic geometry.** Local (L) and Local (E) show that hyperbolic representation itself already brings gains in most of the datasets; while they are not always the best-performing variants, they remain competitive, and the remaining gap is mainly due to the additional benefit of the decoupling strategy.
>
> *  **Role of parameter strategy.** The degradation of w/o DS confirms that parameter decoupling is the key factor, providing significant gains beyond hyperbolic representation alone.
>
> *  **Role of client-specific curvature.** The performance drop of w/o TS confirms the importance of client-specific curvature, while the similar results across different curvature initializations suggest that the key is to let model learn suitable curvature for each client, rather than any specific initializer.
>
> We will make this message explicit in the revised manuscript by reorganizing the ablation discussion into a single, more comprehensive analysis.
>
> &nbsp;
>
> **W2 & Q2: Scalability of FlatLand.**
>
> Thank you for this valuable suggestion. We agree that evaluating under larger-scale settings would further strengthen the paper. In response, we have conducted additional experiments with a substantially larger number of clients, including **50, 100, 200, and 500 clients**, to better assess the scalability of our method under more challenging federated settings.
>
> | Method | 50 | 100 | 200 | 500 |
> | --- | --- | --- | --- | --- |
> | `FedAvg` | 0.559813 | 0.571533 | 0.613157 | 0.663527 |
> | `Fed-Pub` | 0.678136 | 0.706299 | 0.712909 | 0.757341 |
> | `FlatLand(Ours)` | **0.699347** | **0.721644** | **0.742069** | **0.779503** |
>
> These additional results show that **our method consistently maintains its advantage as the number of clients increases, demonstrating its robustness and effectiveness in large-client scenarios**. We will include these results in the revised version and discuss them more explicitly to better highlight the scalability of the proposed approach.

---

> > ### Author Rebuttal · Reviewer_rmZV · 2026-04-04
> >
> > All of my issues have been resolved, and I have decided to keep my score.

---

> > > ### Author Response · Authors · 2026-04-04
> > >
> > > We sincerely appreciate your positive feedback and are pleased that our rebuttal has addressed your concerns. We will incorporate the rebuttal into the final version of our manuscript.

---

### Decision · Program_Chairs · 2026-04-30

**Decision:**

Accept (spotlight)

**Comment:**

This paper addresses personalized federated graph learning under client-level structural heterogeneity, where Euclidean methods may fail to capture differences in graph geometry across clients. The authors propose FlatLand, a framework that places each client in a tailored Lorentz hyperbolic space with learnable curvature and separates personalized time-like parameters from shared space-like parameters during aggregation.

Reviewers agree that the paper tackles an important and practical problem with a clear and original geometric formulation (Reviewer rmZV, Reviewer SdJM, Reviewer ACzZ, Reviewer L7WS). They highlight the use of Lorentz space and client-specific curvature to model heterogeneity, the decoupling of personalized and shared parameters, and the generally strong experimental coverage and presentation quality (Reviewer rmZV, Reviewer SdJM, Reviewer L7WS). The main concerns focus on whether the gains are fully disentangled across hyperbolic representation, curvature learning, and parameter decoupling (Reviewer rmZV, Reviewer L7WS), whether the role of the time-like dimension and the proposed geometric mechanism are directly validated enough (Reviewer SdJM, Reviewer ACzZ, Reviewer L7WS), and whether the method’s applicability, scalability, theoretical assumptions, and curvature choices are sufficiently clarified (Reviewer rmZV, Reviewer SdJM, Reviewer ACzZ, Reviewer L7WS). Still, all reviewers indicated that the rebuttal adequately addressed their concerns, and one reviewer explicitly raised the score after rebuttal.

In conclusion, this submission makes a meaningful and novel contribution by introducing a geometric perspective to personalized federated graph learning and supporting it with solid theory, consistent empirical results, and a successful rebuttal. While some questions remain about mechanism validation, scope across graph structures, and further large-scale analysis, these issues do not outweigh the paper’s originality, technical quality, and broad reviewer support. Balancing the conceptual novelty, methodological soundness, and positive post-rebuttal consensus, my final recommendation is accept.